

# 1 Glacier melting and precipitation trends detected by surface area
# 2 changes in Himalayan ponds

Franco Salerno[1,3], Sudeep Thakuri[1,3], Nicolas Guyennon[2], Gaetano Viviano[1], Gianni Tartari[1,3]
[1] National Research Council, Water Research Institute, Brugherio (IRSA -CNR), Italy
[2] National Research Council, Water Research Institute, Roma (IRSA-CNR), Italy
[3] Ev-K2-CNR Committee, Via San Bernardino, 145, Bergamo 24126, Italy
*Correspondence to Franco Salerno (*salerno@irsa.cnr.it*)*
**Abstract.** Climatic time series for high-elevation Himalayan regions are decidedly scarce. Although
glacier shrinkage is now sufficiently well described, the changes in precipitation and temperature at these
elevations are less clear. This contribution shows that the surface area variations of unconnected glacial
ponds, i.e., ponds not directly connected to glaciers, can be considered suitable proxies for detecting
changes in the main hydrological components of the water balance on the south side of Mt. Everest.
Glacier melt and precipitation trends have been inferred by analyzing the surface area variations of ponds
with various degrees of glacial coverage within the basin. In general, unconnected ponds over the last
fifty years (1963-2013 period) have decreased significantly by approximately 10%. We inferred an
increase in precipitation occurred until the mid-1990s followed by a decrease until recent years. Until the
1990s, glacier melt was constant. An increase occurred in the early 2000s, and in the recent years,
contrasting the observed glacier reduction, a declining trend in maximum temperature has decreased the
glacier melt.

## 20 1 Introduction

Meteorological measurements in high-elevation Himalayan regions are scarce due to the harsh
conditions of these environments, which limit the suitable maintenance of weather stations (e.g., Vuille,
2011; Salerno et al., 2015). Consequently, the availability of long series is even more rare (Barry, 2012;
Rangwala and Miller, 2012; Pepin et al., 2015). Generally, gridded and reanalysis meteorological data are
used to overcome this lack of data and can be considered an alternative (e.g., Yao et al., 2012). However,
in these remote environments their use for climate change impact studies at the synoptic scale must be
performed with caution due to the absence of weather stations across the overall region, which limits the
ability to perform land-based evaluations of these products (e.g., Xie et al., 2007). Consequently, the
meager knowledge on how the climate has changed in recent decades in high-elevation Himalayan
regions presents a serious challenge to interpreting the relationships between causes and recently
observed effects on the cryosphere. Although glaciers reduction in the Himalaya is now sufficiently well
described (Bolch et al., 2012; Yao et al., 2012, Kääb, et al., 2012), the manner in which changes in climate
drivers (precipitation and temperature) have influenced the shrinkage and melting processes is less clear
(e.g., Bolch et al., 2012; Salerno et al., 2015), and this lack of understanding is amplified when forecasts
are conducted.
In this context, a substantial body of research has already demonstrated the high sensitivity of lakes
and ponds to climate (e.g., Pham et al., 2008; Williamson et al., 2008; Adrian et al., 2009; Lami et al.,
2010). Some climate-related signals are highly visible and easily measurable in lakes. For example,



climate-driven fluctuations in lake surface areas have been observed in many remote sites. Smol and
Douglas (2007) reported decadal-scale drying of high Arctic ponds due to changes in the ratio of
precipitation to evaporation. Smith et al. (2005), among other authors, found that lakes in areas of
discontinuous permafrost in Alaska and Siberia have disappeared in recent decades. In the Italian Alps,
Salerno et al. (2014a) found that since the 1980s, lower-elevation ponds have experienced surface area
reductions due to increased evaporation/precipitation ratio, while higher-elevation ponds have increased
in size and new ponds have appeared as a consequence of glacial retreat.
In high Asian mountains and in particular in the interior of the Tibetan Plateau, the observed lake
growth since the late 1990s is mainly attributed to increased precipitation and decreased evaporation (Lei
et al., 2014; Song et al., 2015). In contrast, Zhang et al., 2015, attribute the observed increases in lake
surface areas since the 1990s across entire Pamir-Hindu Kush-Karakoram-Himalaya region and the
Tibetan Plateau region to enhanced glacier melting. Wang et al., 2015, reached similar conclusions in a
basin located in the south-central Himalaya. In our opinion, the divergences in the causes leading to the
lake surface area variations in central Asia are due to the types of glacial lakes considered in these studies,
which could be differentiated in relation to some features of glaciers located within their basin.
In general, in high Asian mountains, three types of glacial lakes can be distinguished according to
Ageta et al. (2000) and Salerno et al. (2012): (i) lakes that are not directly connected with glaciers but that
may have a glacier located in their basin (unconnected glacial lakes); (ii) supraglacial lakes, which
develop on the surface of a downstream portion of a glacier; and (iii) proglacial lakes, which are moraine-
dammed lakes that are in contact with the glacier front. Some of these lakes store large quantities of water
and are susceptible to glacial lake outburst floods (GLOFs). Therefore, in the Himalaya, the potential risk
of GLOFs has been, with good reason, widely investigated (e.g., Richardson and Reynolds, 2000; Benn et
al., 2012). Factors controlling the growth of these lakes depend on the glacier features from which they
develop (surface gradient, mass balance, cumulative surface lowering, and surface velocity) (Reynolds,
2000; Quincey et al., 2007; Sakai and Fujita, 2010; Salerno et al., 2012; Sakai, 2012, Thakuri et al.,
2015). The causes of proglacial lake development are decidedly similar, and supraglacial lakes are
potential precursors of these lakes (e.g., Bolch et al., 2008; Salerno et al., 2012; Thakuri et al., 2015).
Their filling and drainage is linked to the supply of meltwater from snow or glacial sources (Benn et al.,
2001; Liu et al., 2015), and the opeping and closure of englacial conduits (Gulley and Benn, 2007).
Therefore, whereas the lake surface area variations of supraglacial and proglacial lakes are strictly related
to glacier dynamics, the variations in unconnected glacial lakes are only influenced by glacier melting and
precipitation. The different water sources make unconnected glacial lakes potential indicators of changes
in the overall water balance components in high-elevation lake basins, such as precipitation, glacier
melting, and evapotranspiration.
An valuable opportunity for a fine-scale investigation on climate-driven fluctuations in lake surface
area is particularly evident on the south slopes of Mt. Everest (Nepal), which is one of the most heavily
glaciated parts of the Himalaya (Scherler et al., 2011). Additionally, this region is also characterized by
the most glacial lakes in the overall Hindu-Kush-Himalaya range (Gardelle et al., 2011), and a twenty-
year series of temperature and precipitation data has recently been reconstructed for these high elevations
(5000 m a.s.l.) (Salerno et al., 2015). Moreover, the reduced dimensions of the water bodies in this region,
which we can define as ponds according to Hamerlik et al. (2013) (a threshold of $2 \cdot 10^4$ m$^2$ exists between
ponds and lakes), make these environments especially susceptible to the effects of climatic changes
because of their relatively low water volumes and high surface area to depth ratios (Buraschi et al., 2005;
Beniston, 2006).
This contribution examines the surface area changes of unconnected glacial ponds on the south side of





Mt. Everest (an example is shown in Figure 1) during the last fifty years to evaluate whether they act as
potential indicators of changes in the main components of the hydrological cycle (precipitation, glacier
melting, and evapotranspiration) at high elevations in the Himalayan range.

## 87    2 Region of investigation.

The current study is focused on the southern Koshi (KO) Basin, which is located in the eastern part of
central Himalaya (CH) (Yao et al., 2012; Thakuri et al., 2014) (Fig. 2). In particular, the region of
investigation is the southern slopes of Mt. Everest in Sagarmatha (Mt. Everest) National Park (SNP)
(27.75° to 28.11° N; 85.98° to 86.51° E) (Fig. 2a) (Amatya et al., 2010; Salerno et al., 2010). The SNP
(1148 km$^2$) is the highest protected area in the world, extending from an elevation of 2845 to 8848 m a.s.l.
(Salerno et al., 2013). Land cover classification shows that almost one-third of the territory is
characterized by temperate glaciers and that less than 10% of the park area is forested (Bajracharya et al.,
2010), mainly with Abies spectabilis and Betula utilis (Bhuju et al., 2010).
The climate is characterized by monsoons, with a prevailing S-N direction (Ichiyanagi et al., 2007).
For the last twenty years at the Pyramid meteorological station (5050 m a.s.l.) (Fig. 2a), the total annual
accumulated precipitation is 446 mm, with a mean annual temperature of -2.45 °C. In total, 90% of the
precipitation is concentrated during June-September. The probability of snowfall during these months is
very low (4%) but reaches 20% at the annual level. Precipitation linearly increases to an elevation of 2500
m and exponentially decreases at higher elevations (Salerno et al., 2015).
Most of the large glaciers are debris-covered, i.e., the ablation zone is partially covered with
supraglacial debris (e.g., Scherler et al., 2011; Bolch et al., 2011; Thakuri et al., 2014). In the SNP, the
glacier surfaces are distributed from approximately 4300 m to above 8000 m a.s.l., with more than 75% of
the glacier surfaces lying between 5000 m and 6500 m a.s.l. The area-weighted mean elevation of the
glaciers is 5720 m a.s.l. in 2011 (Thakuri et al., 2014). These glaciers are identified as summer-
accumulation glaciers that are fed mainly by summer precipitation from the South Asian monsoon system
(Ageta and Fujita, 1996).
Salerno et al. (2012) realized the complete cadaster of all lakes and ponds in the SNP by digitizing
ALOS-08 imagery and assigning each body of water a univocal numerical code (LCN, lake cadaster
number) according to Tartari et al. (1998). They reported a total of 624 lakes in the park, including 17
proglacial lakes, 437 supraglacial lakes, and 170 unconnected lakes. Previous studies revealed that the
areas of proglacial lakes increased on the south slopes of Mt. Everest since the early 1960s (Bolch et al.,
2008; Tartari et al., 2008; Gardelle et al., 2011, Thakuri et al., 2015). Many studies have indicated that the
current moraine-dammed or ice-dammed lakes are the result of coalescence and growth of supraglacial
lakes (e.g., Fujita et al., 2009; Watanabe et al., 2009; Thompson et al., 2012, Salerno et al., 2012). Such
lakes pose a potential threat due to GLOFs. Imja Tsho (Lake) is one of the proglacial lakes in the Everest
region that developed in the early 1960s as small pond and subsequently continuously expanded (Bolch et
al., 2008; Somos-Valenzuela et al., 2014, Fujita et al., 2009; Thakuri et al., 2015).

## 120    3 Data and Methods

### 121    3.1 Climatic data.

The monthly mean of daily maximum, minimum, and mean temperature and montlly comulated
precipitation time series used in this study have been recently reconstructed for the elevation of the
Pyramid Laboratory (5050 m a.s.l.) (Fig. 2) for the 1994-2013 period (Salerno et al., 2015). The potential





evapotranspiration for the period (2003-2013) has been calculated by applying the Jensen and Haise
model (Jensen and Haise, 1963) using the mean air temperature and solar radiation recorded continuously
during this period at Pyramid Laboratory. The Jensen and Haise model is considered to be one of the most
suitable evaporation estimation methods for high elevations (e.g., Gardelle et al., 2011; Salerno et al.,
2012).
To obtain information on climatic trends in the antecedent period (before the 1990s), we used some
regional gridded and reanalysis datasets. We selected the closest grid point to the location of the Pyramid
Laboratory, and all data were aggregated monthly to allow a comparison at the relevant time scale. With
respect to precipitation, we test the monthly correlation between the Pyramid data and the GPCC (Global
Precipitation Climatology Centre), APHRODITE (Asian Precipitation-Highly Resolved Observational
Data Integration Towards Evaluation of Water Resources), Era-Interim reanalysis of the European Centre
for Medium-Range Weather Forecasts (ECMWF), and CRU (Climate Research Unit -Time Series)
datasets. For mean air temperature, we considered the Era-Interim, CRU, GHCN (Global Historical
Climatology Centre), and NCEP-CFS (National Centers for Environmental Prediction- Climate Forecast
System) datasets, whereas for maximum and minimum temperatures, we used the Era-Interim and NCEP-
CFS datasets (details on the gridded and reanalysis products are reported in Table SI1).
**3.2 Pond digitization.**

Pond surface areas were manually identified and digitized using a morphological map from 1963 and
more recent satellite imagery from 1992 to 2013. In total, five intermediate periods (details on data
sources are provided in Table SI2) were considered according to the availability of satellite imagery. We
selected only those ponds present continuously in all these five periods to exclude possible ephemeral
environments.
From 2000 (2000-2013 period), due to a wider availability of satellite imagery in the region for the
last decade, 10 ponds were selected among the pond population considered in the long-term analysis
(1963-2013) to continuously track the inter-annual variations in surface area in the recent years. The
largest ponds, free from cloud cover, and with diverse glacier coverages (from 1% to 32%) within their
basin were favored in the selection (details on data sources used for these lakes are provided in Table
SI3).
The intra-annual variability in pond surface area has been investigated throughout the year 2001
though the availability of 5 cloud-free satellite images from June to December (details on data sources
used for these lakes are provided in Table SI4). The first semester of the year was excluded from the
analysis because many ponds were frozen until April/May. Even in this case, ponds were selected based
on the absence of cloud cover for all images, and the largest lakes with various degrees of glacial
coverage were favored. Thus, 4 lakes with these characteristics were selected, and their intra-annual
variability is traced in Figure 3. Based on Figure 3, we observe a common significant increase in pond
surface area during the summer months, likely due to monsoon precipitation and high glacier melting
rates. The acceleration disappears during the fall. The period from October to December is the best period
to select the satellite images necessary for the inter-annual analysis of pond surface area. In fact, during
these months, the ponds are not yet frozen, the sky is almost free from cloud cover, and, as observed in
Figure 3, the inter-annual analysis is not affected by intra-annual seasonality. Consequently all images for
the inter-annual analysis have been selected from these months (Table SI1; Table SI2).
**3.3 Glacier surface areas and melt.**



Glacier surface areas within the pond basins were derived from the Landsat 8 remote imagery
(October 10, 2013) taken by the Operational Land Imager (OLI) with a resolution of 15 m. The satellite
imagery used to trace the inter-annual variations in glaciers since the early 1960s is reported in Table SI2.
Detailed information of digitization methods are described in Thakuri et al., 2014.
To simulate the daily melting of the glaciers associated with the 10 selected ponds, we used a simple
T-index model (Hock, 2003). This model is able to generate daily melting discharges as a function of
daily air temperature above zero, the glacier elevation bands, and a melt factor (0.0087 m d$^{-1}$ °C$^{-1}$)
provided by Kayastha et al. (2008) from a field study (Glacier AX010) located close to the SNP. The daily
temperature at the mean elevation of each glacier has been computed according to the lapse rates reported
in Salerno et al., 2015.
**3.4 Morphometric parameters.**
Such parameters related to the ponds basin as the area, slope, aspect, and elevation were calculated
thought the Digital Elevation Model (DEM) derived from the ASTER GDEM. The ASTER GDEM tiles
for the Mt. Everest region were downloaded from http://gdem.ersdac.jspacesystems.or.jp. The vertical and
horizontal   accuracy   of   the   GDEM   are   ~20   m   and   ~30   m,   respectively
(www.jspacesystems.or.jp/ersdac/GDEM/E/4.html). We decided to use the ASTER GDEM instead of the
Shuttle Radar Topography Mission (SRTM) DEM considering the higher resolution (30 m and 90 m,
respectively) and the large data gaps of the SRTM DEM in this study area (Bolch et al., 2011).
Furthermore, the ASTER GDEM shows better performance in mountain terrains (Frey et al., 2012).
**3.5 Uncertainty of measurements.**
All of the imagery and maps were co-registered in the same coordinate system of WGS 1984 UTM
Zone 45N. The Landsat scenes were provided in standard terrain-corrected level (Level 1T) with the use
of ground control points (GCPs) and necessary elevation data (https://earthexplorer.usgs.gov). The
ALOS-08 image used here was orthorectified and corrected for atmospheric effects in Salerno et al.
(2012).

Concerning the accuracy of the measurements, we refer mainly to the work of Tartari et al. (2008),
Salerno et al. (2012), and Salerno et al. (2014a) which address in detail the problem of uncertainty in the
morphologicalal measurements related to ponds and glaciers obtained from remote sensing imagery, maps
and photos. The uncertainty in the measurement of a shape's dimension is dependent both upon the Linear
Error (LE) and its perimeter. In particular for ponds (as discussed also by Fujita et al. (2009), and
Gardelle et al. (2011) in the calculation of LE), only the Linear Resolution Error (LRE) needs to be
considered as the co-registration error does not play a key role. For instance, the ponds considered here
are small, and comparisons are made at the entity level and not at the pixel level. The LRE is limited by
the resolution of the source data. In the specific study of temporal variations of ponds, Fujita et al. (2009)
and Salerno et al. (2012) assumed an error of ±0.5 pixels.
**3.6 Statistical analysis.**
The degree of correlation among the data was verified through the Pearson correlation coefficient (r)
after testing that the quantile-quantile plot of model residuals follows a normal distribution (not shown
here) (e.g., Venables and Ripley, 2002). All tests are implemented in the software R with the significance



level at p <0.05. The normality of the data is tested using the Shapiro–Wilk test (Shapiro and Wilk, 1965;
Hervé, 2015). The data were also tested for homogeneity of variance with the Levene's test (Fox and
Weisberg, 2011). All comparisons conducted in this study are homoscedastic. We used the paired t-test to
compare the means of two normally distributed series. If the series were not normal, as a non-parametric
ANOVA, we used the Friedman test for paired comparisons and the post-hoc test according to Nemenyi
(Pohlert, 2014), while for non-paired comparisons we applied the Kruskal-Wallis test and the post-hoc
test according to Nemenyi-Damico-Wolfe-Dunn (Hothorn et al., 2015). The significance of the temporal
trends has been tested using the Mann Kendall test (p <0.10) (Mann, 1945; Kendall, 1975; Guyennon et
al., 2013).
We conducted a Principal Component Analysis (PCA) as described in Wold et al. (1987) between
pond surface area variations and climatic variables to obtain information on relationships among the data
and to look for reasons that could justify the observed changes in the ponds size (e.g., Settle et al., 2007;
Salerno et al., 2014a,b; Viviano et al., 2014).

### 4 Results and Discussion

### 4.1 Climate reconstruction.

To reconstruct the climatic trends before the 1990s, we compared the annual and seasonal
precipitation and temperature time series recorded at Pyramid station since 1994 (Salerno et al., 2015)
with selected regional gridded and reanalysis datasets (Table SI1). Table 1 shows the coefficient of
correlation found for these comparisons. Era Interim (r = 0.92, p<0.001) for mean temperature (Fig. 4a)
and GPCC (r = 0.92, p<0.001) for precipitation (Fig. 4b) provide the best performance at the annual level.
All these comparisons are shown in Figure SI1. We observe that precipitation increased significantly until
the middle 1990s (+25.6%, p< 0.05, 1970-1995 period), then it started to decrease significantly (-23.9%,
p< 0.01, 1996-2010 period), as observed by the Pyramid station and described by Salerno et al., 2015.
The mean temperature shows a continuous increasing trend (+0.039 °C yr$^{-1}$, p< 0.001. 1979-2013 period)
that has accelerated since the early of 1990s.
Furthermore, Table 1 shows the low capability of all the products to correctly simulate monsoon
temperatures and in particular the daily maximum ones. Figure SI2a reports these correlations at monthly
level, while Figure SI2b highlights the misfit between the temperature trends during the monsoon period.

### 4.2 Pond and glacier surface area variations.

Among the 170 unconnected ponds inventoried in the 2008 satellite imagery (Salerno et al., 2012), we
selected, according to the criteria described above, a total of 64 ponds (approximately 1/3) (Fig. 2a). Table
2 provides a general summary of their morphological features. We prefer to use the median values to
describe these environments because, in general, we observed that these morphological data do not follow
a normal distribution. The population consists of ponds larger than approximately 1 hectare (1.1 10$^4$ m$^2$),
located on very steep slopes (27°), and mainly oriented toward the south-southeast (159°). These ponds
are located at a median elevation of 5181 m a.s.l. and within an elevation zone ranging from 4460 to 5484
m a.s.l..
The observed changes in the surface area of all the considered ponds are listed in Table 3. In general,
all unconnected ponds in the last fifty years (1963-2013) decreased by approximately 10%, with a
significant difference based on the Friedman test (p<0.01). Figure 4d and Table 3 show that, until the
2000s, the ponds had a slight but not significant increasing trend (+7±4%, p>0.05). Since 2000, they have





decreased significantly (-1.7±0.6% yr$^{-1}$, p<0.001 corresponding to -22±18%).

As for glaciers, Figure 4c reports the glaciers surface differences observed across the SNP
(approximately 400 km$^2$) observed by Thakuri et al., 2014. They reported a decrease of -13±3% from
1963 to 2011. We updated this series to 2013 and found a further loss of surface area (-18±3%). For the
glaciers located in the basins with the ponds, we traced changes little bit larger. Their overall surface was
32.2 km$^2$ in 1963 and 25.0 km$^2$ in 2013, with a decrease of -26±20% (Fig. 4c; Table 3). According to
many authors (e.g., Loibl et al., 2014), as we observe here, the main losses in area over the last decades in
the Himalaya have been observed in smaller glaciers.

Once we have analyzed how climate and glacier surface areas have changed over the last fifty years,
we can now attempt to understand the causes that have led to the variations observed in the pond
population. Usually and intuitively, an increase in glacier melt is associated with a decrease in glacier
surface area, as observed here. However, if this inbound component was the most significant element of
the water balance, the ponds would be increased. However, the ponds have decreased since 2000s; thus,
the weaker precipitation observed in recent decades seems to have played a more determining role.
Nonetheless, this analysis is extremely broad because it does not consider, for example, a possible
different relationship between pond surface area and the degree of glacier coverage in the basin.
Therefore, a deeper analysis has been carried out, as shown in the following, to annually trace the surface
areas of 10 selected ponds from 2000 to 2013.

### 265   4.3 Analysis of potential drivers of change.

Table 4 provides the morphometric characteristics of 10 selected ponds. We observe that the median
features of these ponds are comparable with the entire pond population (Table 2), highlighting the good
representativeness of the selected case studies. Figure SI3 shows, for each pond, the annual surface area
variations that occurred during the 2000-2013 period. All the selected ponds show a significant (p<0.05)
decreasing trend according to what has been observed for the whole pond population during the same
period (Table 3; Fig. 4).

These continuous annual series have been compared with temperature (daily maximum, minimum and
mean), precipitation, potential evaporation, and glacier melt of the pre-monsoon, monsoon (Fig. 5), and
post-monsoon seasons. All these trends are noted in Figure SI4, and a correlation table comparing pond
surface area variations and potential drivers of change is presented in Table SI5. In general, we observe
from this table that the highest correlations are found for the monsoon period. The reason is because 90%
of the precipitation and the highest temperatures are recorded during this period (Salerno et al., 2015).
Consequently, the main hydrological processes in the Himalaya occur during the monsoon season.
Focusing on this season, we first observe a large and significant precipitation decrease (-11 mm yr$^{-1}$;
p<0.1) (Fig. 6a; Fig. SI4). Even the mean temperature decreases, but slightly and not significantly (Fig.
SI4). This is a result of a significant decrease in maximum temperature (-0.08 °C yr$^{-1}$; p<0.05) (Fig. 6b;
Fig. SI4) balanced by an increase in minimum temperature (Fig. SI4). The potential evaporation,
calculated on the basis of the mean temperature and global radiation, is constant during the summer
period. These trends have been more broadly discussed in Salerno et al., 2015. These authors, for a longer
period (since 1994), observed that the mean air temperature has increased by 0.9 °C (p<0.05) at the
annual level but that warming has occurred mainly outside the monsoon period and mainly in the
minimum temperatures. Moreover, as we observed here for the last decade, a decrease in maximum
temperature from June to August (-0.05 °C yr$^{-1}$, p<0.1) has been observed. In terms of precipitation, a
substantial reduction during the monsoon season (47%, p<0.05) has been observed.





The glacier melt related to each glacier within the pond basins has been calculated considering both
the both maximum and mean daily temperatures. The averages for all selected cases are analyzed for each
season in Figure SI4, which reveals that the only period producing a sensible contribution is the monsoon
period if the maximum daily temperatures are considered the main driver of the process. The reason can
be easily observed in Figure 2b, which shows the 0 °C isotherms corresponding to the mean and
maximum temperatures. Only the 0 °C isotherm related to the daily maximum temperature during the
monsoon period is located higher than mean elevation of the analyzed glaciers. The T-index model only
calculates the melting associated with temperatures above 0 °C, thereby explaining this pattern. In other
words, the diurnal temperatures influence the melting processes much more the nocturnal ones, which are
considered in the mean daily temperature. Figure 6b shows that the trend is significantly decreasing (3%
$yr^{-1}$, $p<0.05$), according to the decrease observed in maximum temperature.
As anticipated, the highest correlations between ponds surface areas and potential drivers are found
for the monsoon period. Based on Table SI5, we observe that precipitation, maximum monsoon
temperature, and relevant glacier melt are the main drivers of change. The PCA shown in Figure 5
attempts to provide an overall overview of the relationships, during the monsoon period, among the trends
related to the potential drivers of change and the pond surface areas. This representation helps to further
summarize the main components of the water balance system that influence the pond surface areas, i.e.,
glacier melt and precipitation. We observe that evaporation is not a sensible factor at these elevation and
that the evaporation/precipitation ratio is approximately 0.41. Therefore, a hypothetical variation in the
precipitation regime affects the pond water balance two and half times more than the same variation in the
evaporation rate. Moreover, from Figure 5, we observe that there are some ponds that are more correlated
with the monsoon precipitation (i.e., LCN76, LCN141, LCN77, LCN11, and LCN93) and others that are
more correlated with the glacier melt (i.e., LCN68, LCN3, and LCN9). A few ponds seem influenced by
both drivers (i.e., LCN24 and LCN139). The coefficients of correlation are reported in Table 4. According
to the grouping observed with the PCA, Figure 6 shows good fits between the pond surface area trends
and the main drivers of change. Based on Table 4, ponds with higher glacier coverage within the basis
show higher correlations with the glacier melt, and, in contrast, ponds with lower glacier coverage show
higher correlations with precipitation. In our case study, the threshold between the two groups appears to
be a glacier coverage of 10%.

**4.4 Analysis of ponds surface area in the last fifty years.**

Based on the findings related to the main drivers of changes that have influenced the 10 selected
ponds, the overall pond population (all 64 ponds) has been subdivided into two classes defined in relation
to the glacier cover (%) in their basins. In 2013, 25 ponds presented a glacier cover > 10% (i.e., 40% of
the total ponds), and 39 ponds (i.e., 60% of the total ponds) featured glacier coverages less than this
threshold. Hereafter, we define these ponds as ponds without glaciers in the basin (ponds-without-
glaciers), neglecting in this way relatively small glacier bodies, which could possibly be confused with
snowfields. The opposite class is defined as ponds with glaciers in the basin (ponds-with-glaciers).
Among ponds-with-glaciers, Table 2 shows that they are characterized by a median glacier coverage of
19%, oriented toward the east-southeast and very steep (31°).
The observed changes according to this new classification are reported in Table 3. The maps in Figure
7 show the spatial differences between the two classes and comparing the relative annual rate of change,
whereas Figure 8 traces their trends over time. We have already discussed (Fig. 4) that, in general, all
unconnected ponds over the last fifty years have decreased by approximately 10%. Additionally, the



presence of glaciers within the pond basins results in divergent trends. The surface area of ponds-without-
glaciers, from 1963 to 2013, strongly decreased (-25±6%, p<0.001), whereas, for the same period, the
surface area of ponds-with-glaciers decreased much less (-6±2%, p<0.05). Differences in behavior are
also noticeable during the intermediate periods. In this case, we compare the median values of the relative
annual rates of change. From 1963 to 1992, ponds-without-glaciers increased slightly (0.9±0.5% yr$^{-1}$,
p<0.1), whereas the other ones remained constant (0.0 ±0.1% yr$^{-1}$). From 1992 to 2000, ponds-without-
glaciers decreased slightly (-1.1±1.9% yr$^{-1}$, p>0.1), whereas the other ones increased slightly but
significantly (+0.7±0.5% yr$^{-1}$, p<0.05). In the most recent period (2000 to 2013), both categories
decreased, but ponds-without-glaciers decreased more (-2.3±0.7% yr$^{-1}$, p<0.001; -1.5±0.4% yr$^{-1}$,
p<0.001).
The significance of the divergent trend observed between the two groups has been tested for two
periods (1963-1992 and 1992-2013). Ponds-without-glaciers featured significantly (p<0.01) higher
increases than ponds-with-glaciers in the first period (+13±12%; 0±3%, respectively) and significantly
(p<0.01) higher decreases in the second period (-38±6%; -6±2%, respectively), based on a Kruskal-Wallis
test.
**4.5 Change in ponds surface area versus morphological boundary conditions.**
We also analyzed whether ponds belonging to the two classes experienced changes in surface area in
relation to certain morphological boundary conditions, such as the aspect or elevation of the basin. In this
case, we apply the Kruskal-Wallis test as the relevant post-hoc test described above. Figure 9 shows the
surface area changes observed during the 1992-2013 period. The changes were independent of both
elevation and aspect for ponds-without-glaciers (Fig. 9a; Fig. 9c), whereas significant differences can be
observed for ponds-with-glaciers. Ponds located at higher elevations experienced greater decreases (Fig.
9b). In particular, ponds over 5400 m a.s.l. decreased significantly (p<0.01) more than ponds located
below 5100 m a.s.l. In terms of aspect, the south-oriented ponds (Fig. 9d) experienced greater decreases,
which was significantly different from southeast (p<0.01) and southwest (p<0.01) orientations.
The tracing of pond surface areas provides furthermore information on precipitation and glacier melt
trends in space. The decline of precipitation in the SNP since 1992 occurred homogeneously at all
elevations and in all valleys independent of their orientation. Based on the greater loss of surface area for
ponds-with-glaciers at lower elevations, we can infer that glacier melt is higher at these elevations, surely
due to the effect of higher temperatures. Even in valleys oriented in directions other than south, we
observe greater losses in surface area for ponds-with-glaciers. Small glaciers lying in perpendicular
valleys, which are much steeper than the north-south-oriented valleys (following the monsoon direction),
are likely melting more due to their small size and higher gravitational stresses (e.g., Bolch et al., 2008;
Quincey et al., 2009).
**Conclusion**
In high-elevation Himalayan areas, glacial ponds have demonstrated a high sensitivity to climate
change. In general, over the last fifty years, unconnected ponds have decreased significantly by
approximately 10%. We attribute this change to both a drop in precipitation and a decrease in glacier melt
caused by a decline in the maximum temperature in the recent years. The continued shrinkage of glaciers
likely due to the effects of less precipitation than an increase in temperature. Evapotranspiration has little
effect at these elevations and has remained constant over the last decade, during which the main decline
in ponds surface area has been observed.





However, the main contribution provided by this study is to have demonstrated for our case study that
pond surface areas could be traced to detect the behavior of precipitation and glacier melt in remote and
barely accessible regions where, even for recent decades, few or no time series exist. Unfortunately,
before the 2000s, the availability of high-resolution satellite imagery is very limited. However, with the
limited data at our disposal, important information on the evolution of certain main components of the
hydrological cycle at high elevations has been discerned: an increase in precipitation occurred until the
middle 1990s followed by a decrease until recently. Until the 1990s, the glacier melt was constant. Then,
an increase occurred in the early 2000s. In recent years, the declining trend observed for maximum
temperature has reduced the glacier melt.
We observed that simply tracing the glacier surface areas did not yield information on the temporal
behavior of glacier melt. In this regard, a decrease in glacier surface area has been identified over the last
fifty years, but this reduction does not correspond to an increase in glacier melt, as normally expected. As
discussed by other authors (Thakuri et al., 2014; Salerno et al., 2015; Wagnon et al., 2013), on the south
slopes of Mt. Everest, the weaker precipitation is the main cause of glacier shrinkage. In recent years,
glaciers are accumulating less than they were decades ago; thus, their size is declining. In contrast, the
tracing of pond surface areas demonstrates that glacier melt has not increased during the same period. In
fact, the increase in mean air temperature here occurred mainly outside of the summer months and mainly
during the night (Salerno et al., 2015).
Consequently, a question arises in regard to the portability of this method. Here, portability refers to
the degree to which the proposed method is replicable in other remote environments. In the Himalaya,
other land based climatic series at high elevations are decidedly scarce (Barry, 2012; Rangwala and
Miller, 2012; Pepin et al., 2015; Salerno et al., 2015). This constraint limits the ability to further test the
ability of glacier ponds to detect the main water balance components in other Himalayan high-elevation
regions. Therefore, the inferences developed here could be simply applied and trends in precipitation and
glacier melt inferred for the overall mountain range. Observing differences in the magnitude of changes
between the two classes that differ in glacier coverage (threshold of 10%) across different periods, along
an elevation gradient, or according to the basin aspect, as carried out here, could improve the confidence
of the inferred findings. In contrast, in other mountain ranges with other the climatic conditions, the
inferences developed here might not be valid, and station-observed climatic data would be required to test
the ability of glacier ponds to detect the main water balance components.

**Author contributions**

F.S. and G.T. designed research; F.S. N.G. and S.T. analyzed data; F.S. wrote the paper. F.S. N.G. S.T.
G.V. and G.T. data quality check.

**Acknowledgements**

This work was supported by the MIUR through Ev-K2-CNR/SHARE and CNR-DTA/NEXTDATA
project within the framework of the Ev-K2-CNR and Nepal Academy of Science and Technology
(NAST).

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



**Table 1.** Coefficients of correlation between precipitation and temperature time series recorded at
Pyramid station for the 1994-2013 period and gridded and reanalysis datasets (pre-monsoon, monsoon,
and post-monsoon seasons as the months of February to May, June to September, and October to January,
respectively). Bold values are significant with p<0.01.

| | | APHRODITE | GPCC | CRU | ERA Interim |
|---|---|---|---|---|---|
| **Precipitation** | annual | 0.43 | **0.75** | 0.34 | 0.33 |

| | | NCEP CFS | GHCN | CRU | ERA Interim |
|---|---|---|---|---|---|
| **Minimum Temperature** | pre monsoon | **0.64** | | | **0.81** |
| | monsoon | 0.47 | | | **0.72** |
| | post monsoon | **0.70** | | | **0.65** |
| | annual | **0.72** | | | **0.92** |
| **Mean Temperature** | pre monsoon | **0.79** | **0.83** | **0.8** | **0.87** |
| | monsoon | 0.61 | 0.51 | 0.42 | **0.67** |
| | post monsoon | **0.79** | **0.77** | 0.57 | **0.82** |
| | annual | **0.81** | **0.85** | **0.89** | **0.92** |
| **Maximum Temperature** | pre monsoon | **0.83** | | | **0.88** |
| | monsoon | 0.54 | | | 0.45 |
| | post monsoon | **0.82** | | | **0.86** |
| | annual | **0.70** | | | **0.80** |


















**Table 2.** General summary of the morphological features of all the considered ponds (data from 2013).

| Topography | Glacier cover <5% median (range) | Glacier cover >5% median (range) | All lakes Median median (range) |
|---|---|---|---|
| Pond elevation (m a.s.l.) | 5181(4460-5484) | 5159(4505-5477) | 5170(4460-5484) |
| Pond area ($10^4$ m$^2$) | 0.8(0.1-6.2) | 1.3(0.3-56.3) | 1.1(0.1-56.2) |
| Basin area ($10^4$ m$^2$) | 30(2-430) | 130(30-2300) | 70(2-2300) |
| Basin slope (°) | 25(10-39) | 29(23-41) | 27(10-41) |
| Basin aspect (°) | 163(68-256) | 141(94-280) | 159(68-280) |
| Basin mean elevation (m a.s.l.) | 5293(4760-5531) | 5400(5119-5945) | 5315(4760-5945) |
| Basin/Lake area ratio (m$^2$/m$^2$) | 60(3-485) | 67(10-523) | 64(3-523) |
| Glacier area (%) | 0(0-4) | 19(0-61) | 0.5(0-61) |
| Glacier slope (°) | - | 31(21-38) | - |
| Glacier aspect (°) | - | 124(150-250) | - |
| Glacier mean elevation (m a.s.l.) | - | 5680(5470-7500) | - |





















**Table 3.** General summary of pond surface area changes from 1963 to 2013. The surface area changes of
the glaciers located within the basins are also reported.

| Period | Pond surface area change | | | Glacier surface area change | Period | Pond surface area change | | | |
|---|---|---|---|---|---|---|---|---|---|
| | Cumulative loss (%) | | | Cumulative loss (%) | | Relative annual rate (% yr$^{-1}$) | | | |
| Glacier coverage | < 5% | > 5% | All ponds | All basins | Glacier coverage | < 5% | > 5% | | All ponds |
| **1963-1992** | **+13±12** · | **0 ±3** | **+3 ±7** | **8 ±8** | **1963-1992** | **0.9 ±0.5** · | **0.0 ±0.1** | | **+0.5 ±0.3** |
| **1963-2000** | -1 ±6 | +9 ±2 * | +7 ±4 | -2 ±8 | **1992-2000** | **-1.1 ±1.9** | **+0.7 ±0.5** · | | **-0.4 ± 0.1** |
| **1963-2008** | -4 ±5 | +3 ±2 | +1 ±4 | -13 ±9 ** | 2000-2008 | -0.3 ±1.0 | -1.6±0.6 | | -0.7 ± 0.7 |
| **1963-2011** | -7 ±6 | 0 ±2 | -2 ±5 | -14 ±14 ** | 2008-2011 | 0.0 ±2.8 | 0.0 ±1.6 | | 0.0 ±2.2 |
| **1963-2013** | **-25 ±6 *** | **-6 ±2 *** | **-10 ±5 *** | **-26 ±20 *** | 2011-2013 | -12.9 ±4.4 *** | -5.8 ±2.5 * | | -11 ±3.5** |
| **1992-2013** | **-38 ±6 *** | **-6 ±2 *** | **-13 ±5 *** | **-34 ±15 *** | 2000-2013 | **-2.3±0.7** ··· | **-1.5 ±0.4** ··· | | **-1.7 ±0.6** ··· |






















**Table 4.** Morphometric features of 10 selected ponds considered in the 2000-2013 analysis. Data are from 2013. Coefficients of correlation are for the monsoon season. The relationships with the other seasons are reported in Table SI5.

| Pond Code | Glacier Cover (%) | Pond Elevation (m a.s.l.) | Basin Aspect (°) | Basin Slope (°) | Basin Area (km²) | Pond Area (10⁴ m²) | Basin Elevation (m a.s.l.) | Coefficient of Correlation (Ponds surface area vs Precipitation) | Coefficient of Correlation (Ponds surface area vs Glacier melt) |
|---|---|---|---|---|---|---|---|---|---|
| LCN139 | 1 | 4749 | 75 | 30 | 0.6 | 4.6 | 5596 | 0.50 | 0.35 |
| LCN93 | 2 | 5244 | 116 | 23 | 0.7 | 0.6 | 5502 | 0.70 ** | 0.39 |
| LCN141 | 3 | 5316 | 152 | 27 | 1.4 | 2.6 | 5701 | 0.72 ** | 0.37 |
| LCN11 | 3 | 5029 | 229 | 24 | 1.2 | 1.8 | 5372 | 0.76 ** | 0.49 |
| LCN77 | 7 | 4920 | 142 | 26 | 8.6 | 18.3 | 5507 | 0.55 * | 0.29 |
| LCN76 | 9 | 4800 | 140 | 25 | 13.6 | 59.2 | 5457 | 0.65 ** | 0.23 |
| LCN24 | 10 | 4466 | 162 | 28 | 23.0 | 54.0 | 5477 | 0.44 | 0.65 ** |
| LCN9 | 13 | 5202 | 117 | 36 | 0.7 | 0.6 | 5792 | -0.27 | 0.61 ** |
| LCN3 | 30 | 5261 | 154 | 35 | 2.0 | 11.7 | 5981 | 0.17 | 0.87 *** |
| LCN68 | 32 | 5006 | 232 | 35 | 1.2 | 3.2 | 5686 | 0.12 | 0.65 ** |
| **Median** | 8 | 5018 | 147 | 28 | 1.3 | 3.9 | 5551 | | |

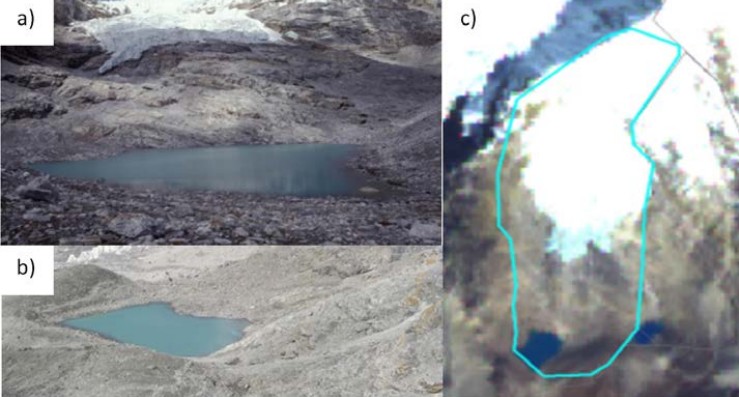

**Figure 1**. Example of an unconnected glacial pond (LCN5) with a glacier within the basin. Pictures were taken in September 1992: a) view looking north showing the distance between the glacier and the pond surface; b) from east showing the frontal moraine. c) LCN5 basin tracked on ALOS 2008 imagery.






















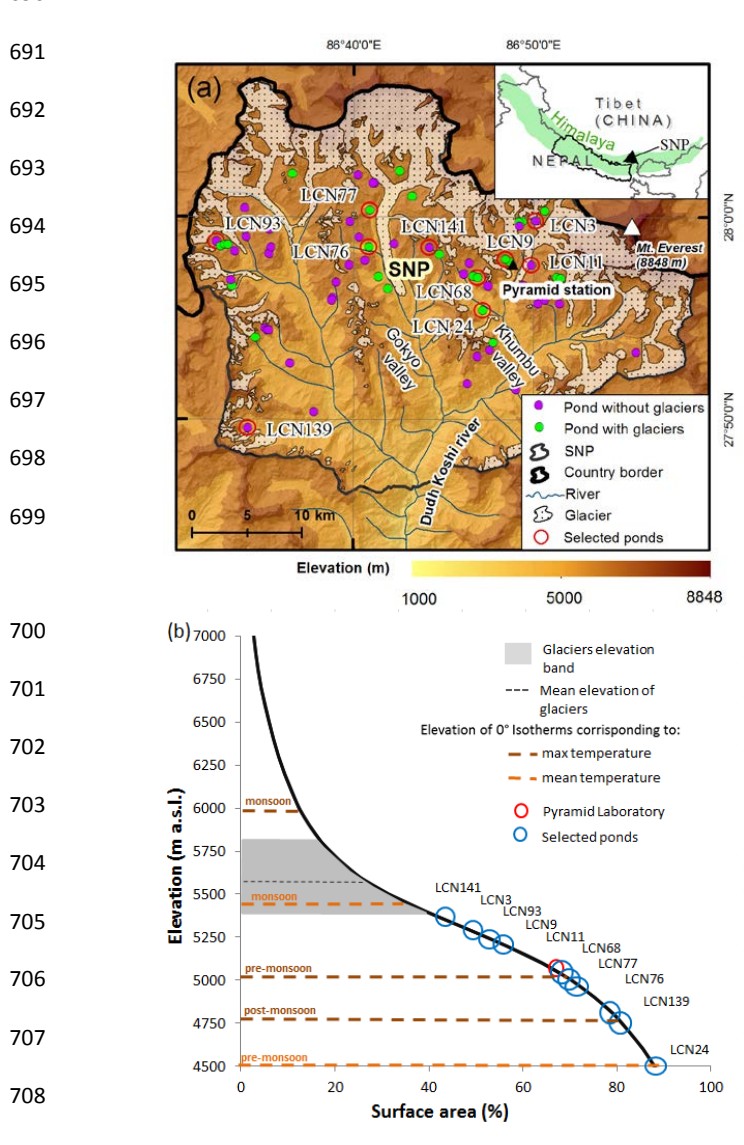

**Figure 2.** a) Location of the study area in the Himalaya and a detailed map of the spatial distribution of
all unconnected ponds analyzed in this study. b) Hypsometric curve of SNP. Along this curve, the
locations of 10 selected ponds are shown. The 0 °C isotherms corresponding to the mean and maximum
temperature in 2013 are plotted for the pre-, post-, and monsoon period according to the lapse rates
reported in Salerno et al., 2015. The mean glacier elevation distribution (mean ± 1 standard deviation) of
10 selected ponds and the location of the…Pyramid meteorological station are also reported.



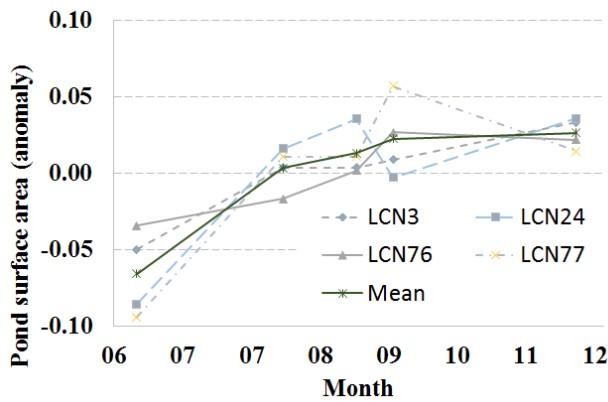

**Figure 3.** Intra-annual analysis (June-December) of selected pond surface areas



















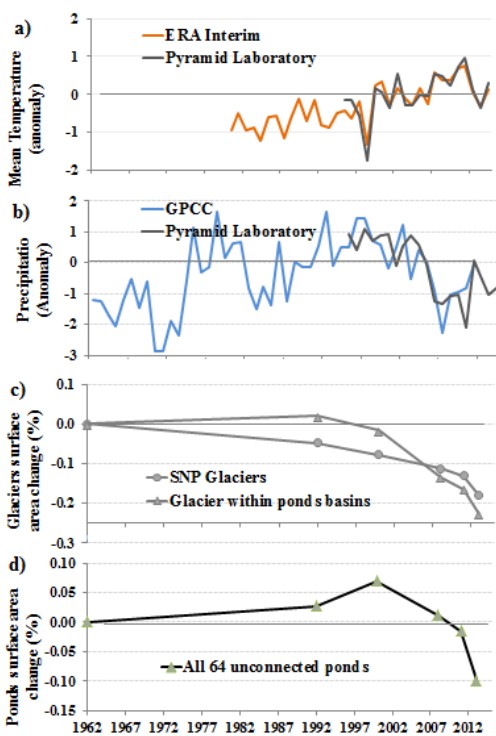


**Figure 4.** Trend analysis of climate, glacier area and ponds surface area for the last fifty years in the SNP.
a) and b) Trends are expressed in terms of anomalies with respect to the mean value calculated for the
considered period. c) and d) The relative variations with respect to 1963 are represented. a) Era Interim
mean annual temperature compared with Pyramid's land-based data. b) GPCC annual precipitation and
Pyramid's land-based data. c) Glacier surface area variations for the overall SNP (Thakuri et al., 2014)
and for glaciers located in basins with ponds. d) Surface area variations of the ponds.












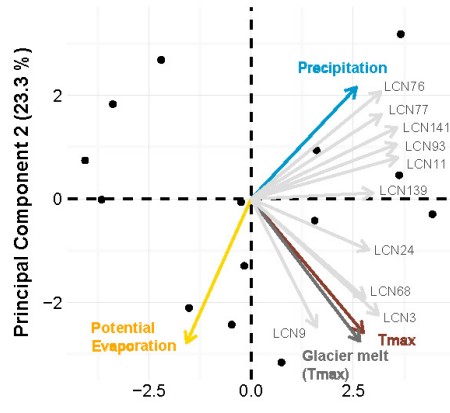


**Figure 5.** Principal Component Analyses (PCAs) between pond surface area from 2000 to 2013 and potential drivers of change (maximum temperature, precipitation, glacier melt, and potential evaporation) related to the monsoon season. Coefficients of correlation are reported in Table SI5. All trends related to ponds and variables are provided in Figure SI1 and SI2.


















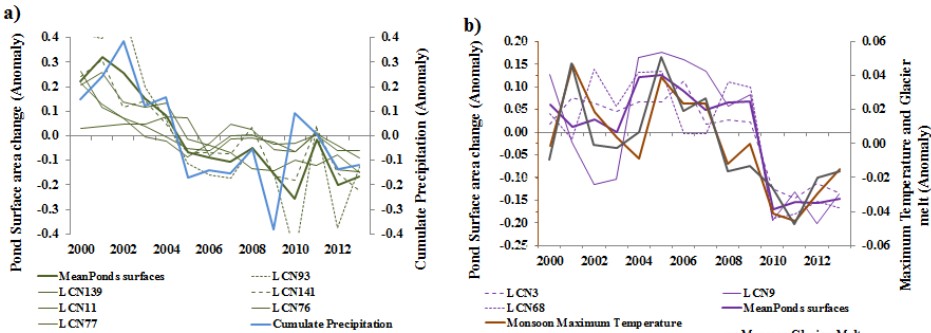

**Figure 6**. Annual trends from 2000 to 2013 related to pond surface area grouped according to the relevant main drivers of change (monsoon season): a) glacier melt (maximum temperature), b) precipitation. Coefficients of correlation are reported in Table SI5. All trends related to ponds and variables are provided in Figure SI1 and SI2.





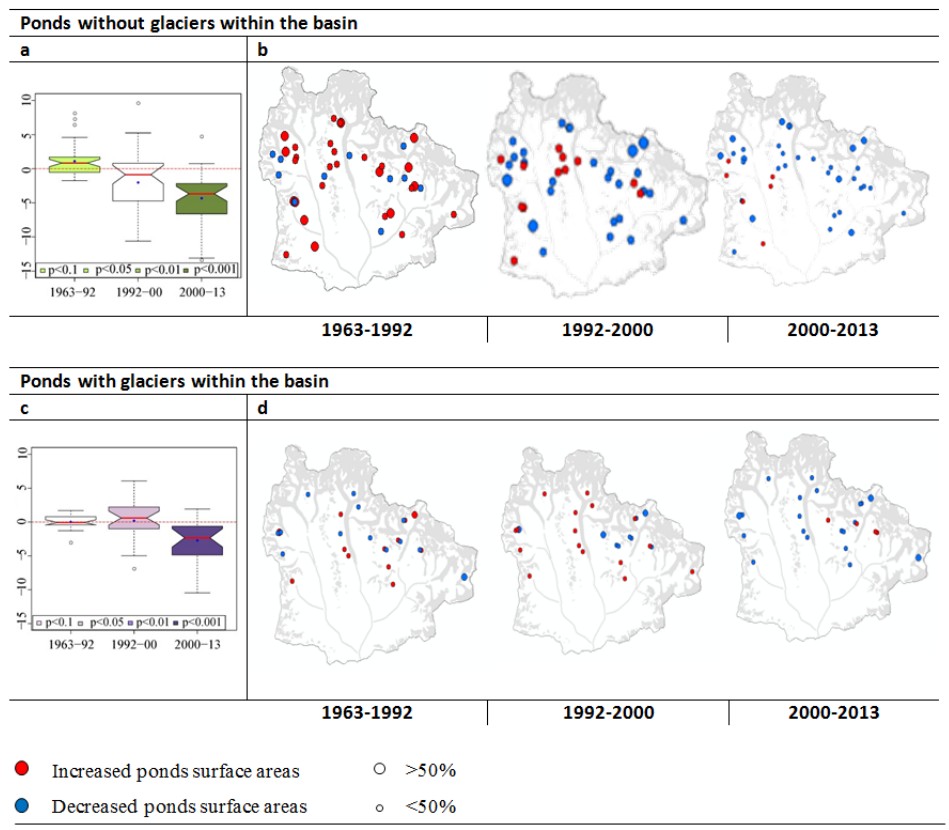

**Figure 7**. Changes in pond surface area in the Mt. Everest region. The left boxplots represent the annual rates of change of ponds in the analyzed periods: (a) ponds with glaciers within the basin, (c) ponds without glaciers within the basin. The blue points in the boxplots indicate the mean, whereas the red line is the median. On the right side, the maps (b, d) visualize the variations that occurred in the pond population during the same three periods considered in the relevant boxplots on the left. Reference data are reported in Table 3. All percentages refer to the initial year of the analysis (1963).



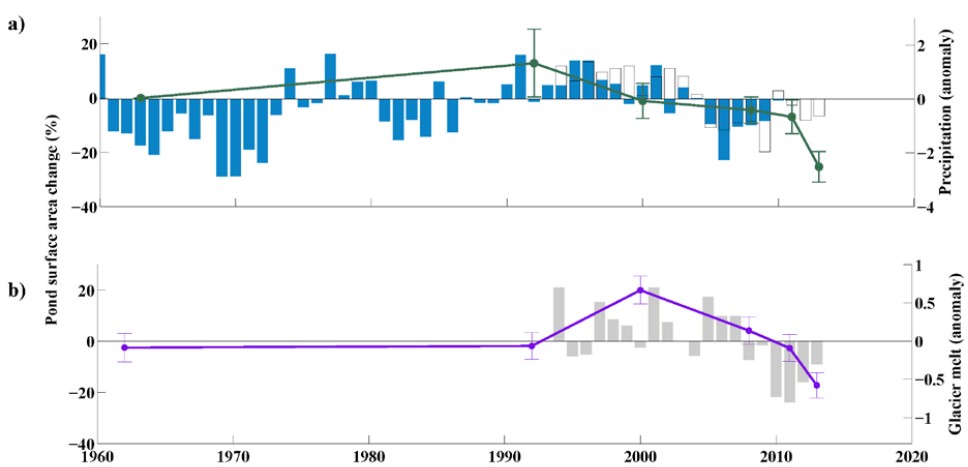


**Figure 8**. Trend analysis for the last fifty years of pond surface area in the SNP for a) ponds-without-
glaciers and b) ponds-with-glaciers.











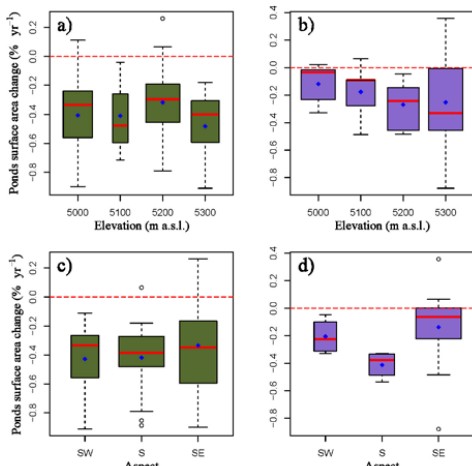


**Figure 9**. Pond surface area changes observed during the 1992-2013 period in relation to certain
morphological boundary conditions in the basin: elevation (upper graphs) and aspect (lower graphs). On
the left ponds-without-glaciers, and on the right ponds-with-glaciers.
