# Peer review of "Glacier melting and precipitation trends detected by surface area 2 changes in Himalayan ponds"

_The Cryosphere, 2016_

## Referee Comment (RC1) · Anonymous Referee #1 · 28 Apr 2016

General comments

Understanding the links between climate, glaciers and hydrology in high mountain area is a growing and very important topic. This paper builds on other work by this group. There is potentially an interesting paper in here, which is novel and might lead the way to demonstrating how the changing size of ponds in mountainous regions that are not in immediate contact with ice but which contain glaciers in their catchments might be used to infer spatial and temporal trends in climate (precipitation, temperature, evaporation, glacier melt). The paper uses a statistical approach to the problem and the authors are to be commended for such a detailed analysis. Eventually one might imagine being able to use perhaps a more robust physically based approach, similar to that used by, e.g., Leclercq & Oerlemans, to reconstruct climate from glacier length fluctuations. This paper could be a useful stepping stone in that direction.

[Figure]

[P.W. Leclercq, J. Oerlemans 2012. Global and hemispheric temperature reconstruction from glacier length fluctuations Climate Dynamics 38 1065-1079, doi: 10.1007/s00382-011-1145-7]

I see 4 key problems with the paper as it currently stands although I hope the authors might be able to deal with these, re-orientate, focus, correct things and rewrite the paper so that it provides a better contribution to the cryospheric sciences.

1. The aim, objectives and overall general methodology of the paper are not articulated towards the beginning of the paper, so that the reader [or this one at least] remains generally confused about what is being done and, more importantly, why things are being done and has to gradually piece things together while reading the paper.

2. The paper is very involved and dense with lots of different levels of analyses, and lacks a clear focus of what it is trying to achieve. I'd encourage the authors to work out what the key take home messages of the paper are and to present only the material that leads to those conclusions.

3. The paper is hard to follow, with sufficient ambiguities, inconsistencies, apparent contradictions and small lapses in grammar and syntax, to justify rewriting quite large sections, especially the Abstract and Conclusions. It would benefit from running through a spell checker and from proof reading by a native English speaker if at all possible.

4. I query some of the scientific assumptions / results

I elaborate on these points below.

Specific comments

1. The paper needs to articulate what the overall aims, objectives and methodology are. Currently, all we have on lines 83-86 is this:

"This contribution examines the surface area changes of unconnected glacial ponds

on the south side of Mt. Everest (an example is shown in Figure 1) during the last fifty years to evaluate whether they act as potential indicators of changes in the main components of the hydrological cycle (precipitation, glacier melting, and evapotranspiration) at high elevations in the Himalayan range." Even as a general aim, this is rather vague. This needs tightening up, we need to be given some more specific objectives and told an overall methodology of how these objectives will be achieved. Currently, after these 5 lines, we have an introduction to the field area (Section 2) followed by a detailed section on Data and Methods (Section 3). But when reading Section 3, we don't know why we're being told about the climate data, digitization of ponds, calculation of glacier surface area and melt, derivation of morphological parameters , etc.

For example, on line 203 you refer to "degree of correlation among the data" But we have no idea what precise data you're talking about, nor why you want to correlate them.

2. The paper is very detailed, convoluted and involved, with a lot of separate components: i) looking at correlations between reanalysis climate data and ground climate data after 1994 to see which reanalysis products may most reliably be used to infer climate in the region prior to 1994;

ii) generating other proxy data ultimately from the climate data, notably evapotranspiration and glacier melt (using a simple temperature index model);

iii) calculating glacier shrinkage and "unconnected pond" area shrinkage (where "unconnected ponds" refer to those not physically in contact with glacier ice) for 6 time periods since 1963 from a map (1963) and satellite imagery (1992, 2000, 2008, 2011, 2013);

iv) performing a suite of non-parametric statistical tests to investigate whether trends in pond area, glacier area, climate & climate derivatives (evapotranspiration and glacier melt) are statistically significant in different time periods (e.g. the whole period 1963-2013 or sub-periods 1963-1992, 1992-2013); between different types of unconnected

pond (those whose upstream catchment is > 10% or < 10% glacierised) or for different "morphological boundary conditions" (e.g. elevation, aspect);

v) performing a Principal Components Analysis on the variables to investigate climate drivers of pond area change.

Furthermore, some of the analysis is done on the full set of 64 ponds, and some is done on a sub-set of 10 ponds. Similarly, some of the analysis splits the time period into two (1963-1992 and 1992-2013) and some splits the time period into three (1963-1992, 1992-2000 and 2000-2013). All in all, the reader gets rather bogged down in the detailed analysis and loses a sense of the big picture.

3. Because the paper has many different strands, it is particularly important to have a very clear abstract and conclusion. Reading the abstract, it is not at all clear what the key take home messages of the work are. Unfortunately, having ploughed my way through the paper and emerged somewhat exhausted from the final sentence of the conclusions, I was still rather unsure what the key conclusions were.

Lines 369-371 tell us that during the monsoon period the "unconnected ponds" declined in area (by 10%). Fine, this is clear.

Lines 371-372 tell us that this is due to a drop in precipitation and a decrease in maximum temperature (and therefore glacier melt). Also quite clear.

Then it gets confusing. Lines 372-373 tell us that "the continued shrinkage of glaciers likely due to the effects of less precipitation than an increase in temperature". This is not a grammatically correct sentence but I assume the authors mean that "the continued shrinkage of glaciers [is] likely due to the effects of less precipitation [rather] than an increase in temperature." I don't recall where in the paper this was discussed. The paper involved a statistical analysis explaining variation in pond area not glacier area. By "continued shrinkage" I assume the authors are referring to the actual shrinkage that occurred in the past, and are not speculating about shrinkage that may or may

not occur in the near future? Note how we're told that pond area shrinkage is due to a "decrease in maximum temperatures" but that glacier shrinkage is likely not due to an "increase in temperature". It's a little ambiguous whether temperatures have, in fact, increased or decreased over the time period. On line 280 we're told that the mean temperature decreased, although not significantly. On line 281 we're told that maximum temperatures decreased. On line 282 we're told that minimum temperatures increased. Actually we're told that the increase in the minimum temperature "balanced" the decrease in the maximum temperature, although this isn't strictly correct as then, I assume, the mean would stay exactly the same. Is it really the case that mean temperature decreased? Figure 4a, shows that the mean temperature increased over the time period!

Section 4.3 is virtually impossible to follow. It spans just a side of A4 during which we're asked to study Table 4, then Table 2, then Fig SI3, Table 3 and Figure 4. That's just the first short paragraph. We then need to look at Fig 5, SI4 and SI5, Fig 6a and SI4, back to 6b, back to SI4, then again, and again, then flip back to 2b. We then have to jump forward again to 6b, move to Table SI5, Figure 5, and Fig 5 again, Table 4, Figure 6 and finally back to Table 4.

I was concerned throughout this section that I was moving the pages back and forth so much that I'd accidentally end up making some sort of 3D origami animal. I'd encourage the authors to cut down on the Figures and Tables and discuss things in a way that doesn't involve so much movement.

4-1. Can you explain better how melt is being derived for the glaciers? In lines 171-176, is it necessary to refer to the work of Salerno et al (2015) regarding the calculation of temperature at the mean elevation of each glacier? Is it not the case that the pyramid data are used together with a lapse rate (tell us what the lapse rate is) and the melt factor to calculate the melt across each elevation band (tell us what the band width is and what DEM is used) and that these are then summed for each glacier to calculate the melt to each glacier?

4-2. Given the way that you're calculating glacier melt, there will be huge autocorrelation between Tmax and Glacier melt. So it's not surprising that your correlation coefficients involving Tmax and Glacier melt are so similar. I'm therefore surprised by Fig 5 where you seem to show that glacier melt and Tmax are two strong independent variables contributing to the principle components. Have I understood this correctly?

4-3. Table SI5. Do I understand this analysis correctly? For each pond, are you only working with 14 data points? Is this sufficient to demonstrate every variable is normally distributed so that you can use the parametric correlation test (as you state you do lines 203-5)

4-4.k On line 100 you tell us that the precipitation has a specific gradient. Given that you go to all the trouble of calculating glacier melt using a lapse rate, and given the importance of precipitation for your analysis, why do you not use this lapse rate in the calculation of precipitation from the pyramid station when analysing the precipitation relevant to the different ponds? The ponds are at different elevations, and the catchments above them have different elevation ranges (and hypsometries). The pptn gradient above 2500m is non-linear. All these things will mean the precipitation falling above the lakes in your analysis will be very different for the different lakes.

4-5. Section 3.5. I'd like to see a better articulation of the sources of error and how they were calculated for this study. First you imply error is a function of linear error and perimeter. Then you refer to a linear resolution error and a co-registration error. This all needs explaining more carefully and precisely.

Technical corrections; typing errors, etc.

There are a lot and I don't have time to give them all. Below I give some of the key ones. Numbers refer to line numbers.

14. "unconnected ponds" This is defined in the paper but the abstract should be intelligible on its own. Explain what is meant here.

[Figure]

15. "We infer an..."

17-19. Rewrite. I think this should be at least 2 sentences. Meaning not at all clear.

31. glacier

44. "...increases in the evaporation / precipitation ratio..." [refer to evaporation / precipitation ratio also above on line 41 to be consistent]

51-53. Vague. Rewrite.

61. What do you mean by "these lakes"? Just proglacial lakes or all 3 categories?

64. "decidedly similar". To what?

67 opening

67. Ref to englacial conduits is relevant to supraglacial lakes but not proglacial.

54-72. Para could be shorter with tighter articulation of key relevant points.

73 A valuable

75 glacierized not glaciated.

75-6. "...region has the largest number of lakes in..."

78. reduced dimensions. Do you mean "relatively small size"?

80 "...make them especially..."

78-82. This sentence is confusing. Is it their small size that's relevant or the low water volumes and high surface area to depth ratios. You start the sentence implying it's the first, and end saying it's the 2nd & 3rd attribute that's important. Rewrite.

79. Can you check the entire document? Here you define lakes and ponds according to size. But earlier and later you use the terms interchangeably and (according to this definition) sometimes incorrectly. You need consistency. Define at the very start of the

paper. You could use "water bodies" if you want a generic term.

89. Do you need the abbreviation "CH"? Do you use this term again?

93-4. "...of the territory contains temperate glaciers and less than 10% is forested."

97. "For the last 20 years" Avoid phrases like this. Later you refer to "the last decade" I think too. These phrases are ambiguous. The last 20 years means 1996-2016 to me, but actually pyramid station has been operating since 1994. Always state the precise dates to avoid confusion.

99 "...precipitation falls between June and Sept..."

102. "...large glaciers in the SNP are..."

103. Delete "In the SNP"

109. "realised the complete cadaster" What does this mean?

110. "univocal" suggest change to "unique"

113 "...Everest after the..."

118. check grammer here.

122 "...and the monthly cumulated..."

123 delete "recently"

125. Why evapotranspiration not also calculated for 1994-2002?

126. "recorded continuously" Is this a monthly time-series too? Or calculated more frequently and averaged?

130. You casually say "before the 1990s" but you should say before 1994. See other instances of this throughout the paper,

143. "intermediate periods" is confusing. Why not just say "scenes"?

146. "environments" is completely the wrong word. Do you mean "biases?

147-8 "For the 2000 – 2013 period, due to the wider availability of satellite imagery, ten ponds were…"

155. Semester is the wrong word

158. "these characteristics" What characteristics are you talking about here?

161. "The acceleration disappears" This is wrong. No acceleration has been discussed previously. Do you mean that there is a decrease in area?

167. "pond basins" This is a bit unclear. You're referring to the basins (or catchments) containing? Or Upsteam of? The ponds.

178. remove the phrase "such". Just list all the parameters you use.

180-181. Vertical accuracy greater than horizontal? Are you sure?

185. Is this EM also used for defining the elevation bands for the calculation of melt? Should have been referred to earlier.

187. Map not maps.

194. morphological? Or best to use morphometric for consistency.

217 pond size

221 before 1994

223. Why are seasonal data shown for temperature but not precipitation in Table 1?

235. Are the 170 ponds all from the SNP region?

237. delete "prefer to"

238. "environments"? Do you mean ponds? Water bodies?

235-242. You don't refer to columns 1 & 2 in Table 2. Are these redundant? Remove

them?

248. "glacier surface differences" ? Do you mean glacier surface area changes?

250. Further loss of area (-18%) is ambiguous. It's not an extra 18% loss since 2011.

251. Poor grammer

255 "Having analysed..."

257. delete "Usually and"

258. "this inbound component" Do you mean glacier melt input?

259-264. Vague, confusing and poor English here.

302. don't need the word "monsoon" at the end of this line with reference to temperature here do you? All these variables are for the monsoon right?

303 "relevant" is the wrong word

307 "sensible factor" is incorrect.

322 "...ponds were in catchments with a glacier..."

323-3. Needs writing.

324-5. Why are you calling ponds in catchments that are <10% glacierised "ponds without glaciers"? Why not just call them "ponds in catchments that are <10% glacierised"?

336. "during the intermediate periods" is confusing. Do you mean in the 1st, 2nd and 3rd part of the 1963-2013 period?

344 "...glaciers had significantly..."

344-5. Rewrite.

359. "...tracing of pond surface area". The word "tracing is not quite correct" Check entire document as this has been used a few places. The word "measuring" would be

better.

370 and 374. First you talk about "over the last 50 years" and then "over the last decade". Why not first discuss the full conclusions of the long term 1963-2013 analysis and then talk about the full conclusions associated with the 2003-2013 work. As stated earlier, I suggest you avoid these phrases.

394-405. This part of the conclusions seems rather weak and not a good place to end.

Table 2. Lakes & Ponds seem to be used interchangeably here. In the Table heading, explain the 3 columns. And is this the sample of 64 or 10 ponds shown here? Median is written twice in the column 3 heading. And in the final column the maximum area for pond area should read 56.3 not 56.2.

Figure 2a. I may be wrong but I think it's only once we look at this Figure that we learn that some ponds do not have glaciers in them. There are 10 selected ponds on this Figure but in the text referring to it I think you said you selected 64 ponds.

Fig 4c and d. Y axis label should read "fraction" not "%" or the numbers should be multiplied by 100. First data point needs to be plotted against 1963 not 1962!

Fig 7. Blue dots depicting the mean in the box plots are barely legible, esp. in the blue 2000-13 Figure c. Is there some distortion as the circles look like ovals?

Fig 8. Heading is wrong.

Fig 9. Change colour scheme as blue dots are invisible

---

## Short Comment (SC1) · 3 May 2016

This paper reports on the surface area changes of unconnected glacial ponds in the south of Mt Everest during the period 1962 to 2013 using maps and satellite images. This time-series data is analysed to identify the drivers of the change using statistical analysis of the correlations with available meteorological data. However, the present draft may greatly benefit from a more careful analysis of this very interesting data set, and also a slightly more systematic description of the methodological details.

My major concerns are as follows:

1) While it has been argued at the outset rather briefly that lakes and ponds are sensitive indicators of climate change, this point demands more serious consideration. The cited references of Beniston et al, 2006 do not seem to discuss lake/pond, while the

[Figure]

other referred article by Burasachi et al, 2005 does not include a relevant discussion of climate sensitivity of the lake/pond area and particularly of the response time scales.

The temporal variation of the surface area of a given pond must be controlled by 1) the balance between water in and out - therefore by the climate, and 2) the bathymetry. But, any attempt to infer climate signal from sparse point measurements of such a time series has to take into account the relevant time-scales associated with response to the fluctuating climate variables.

For example in figure 3, some of the biggest ponds/lakes (eg LCN77, LCN24) show large ( 5%) increase/decrease in their area in a month's time, indicating a strong control of high frequency changes of the climate variables. For the rest of the ponds which are even smaller in size, these high frequency noise would presumably be even larger. How can this sparse time series with high frequency 'noise' that is of similar magnitude as the low frequency signal ( 10% change over 50 years), possibly be used to infer low frequency changes of the climate? In this context it may be noted that glacier length fluctuations can be inverted for temperature change as their slow response makes them immune to high frequency noise.

Similar large fluctuations are also seen in the annual rates reported in Table 3: During 1992 to 2011, rates are very small or insignificant and then there is a very large (1 to 2 order of magnitude larger than the background) spike during 2011-2013. In fact this spike dominates the mean. Is this a signal from a particular short-lived event picked up due to sparse sampling or a real climate change signal? Surprisingly, no such sharp changes are seen in the precipitation or glacier melt data during 2011-2013 as presented here. This needs to be considered very very carefully before accepting the interpretation offered by the authors here.

Further, this issue of high frequency noise can not be overcome simply by averaging over a large set of ponds from the same region, as they are all seeing a strongly correlated noise due to their spatial proximity. And of course, practical limitations like

unavailability of suitable images etc would prevent a higher temporal resolution.

2) While the authors have employed a careful statistical methods to derive their conclusion regarding the climate signal, some simple physical considerations might strengthen their analysis. For example, the climate data (reanalysis/gridded) used is from the grid point that is closest to the Pyramid station. Would not be better to use the grid point closest to a given pond for analysing the area change data for that particular pond? This choice might have led to serious biases in the results as pointed out below.

All the 'ponds with glaciers' (LCN 24,9,3,68) that show significant correlation with glacier-melt, are located in the Khumbu valley, within may be five km of the Pyramid station. So, how can one be sure about the controlling factor behind this pattern - Is it the glacier cover as claimed, or it is just the proximity of the grid point? In fact, data from LCN11 in the same valley has a relatively large correlation coefficient ( 0.5, though probably not significant) with calculated glacier melt, while far-away 'ponds with glaciers' (LCN 76 and 77) has small (0.2-0.3) correlation with the glacier melt. This requires explanation.

Incidentally, there seem to be some ambiguity regarding the definition of two pond classes: with and without glaciers. Table 3 uses 5% as a threshold; text gives a threshold of 10%; Table 4 says LCN3 has 30% glacier cover, while Figure 3 claims LCN3 is a pond without glacier. These differences need to be clarified and the sensitivity of the conclusions to this choice of threshold value may be discussed.

Also, the authors may discuss the spatial pattern of changes as seen in figure 7. For example, looking at this figure it seems statistically significant differences may emerge in trends from the set of ponds near Ngzumpa glacier (Gokyo valley ) and Khumbu glacier (Khumbu valley), irrespective of glacier cover extent. If so, then what is the relevant control, having more than 5% glacier area or the ponds being in the same valley?

In addition, the ponds with glacier cover seem to be larger (table 2). Could it be that the

difference in shrinkage are correlated with pond size? A possibly larger intrinsic climate sensitivity of the smaller ponds could be an alternate explanation for the differences seen between the signal from the two class. This possibility needs to be ruled out as well to justify the conclusions reached.

Other comments:

1) Many of figures presented needs to be carefully redone, checking the axes labels for missing units, choosing proper x and y range so that all data-points are seen, putting legends that are missing, giving complete and accurate plot captions etc. Some examples: i) what are the units of vertical scales in figure 3, 4a, 4b, 6a, ... ii) Figure 3 horizontal axis: tics read 06,07,07,08,... . Also horizontal separation of the points are inconsistent with time stamps given in table SI4. iii) what are the criteria for the selecting the ponds whose records are presented in figure 6? why LCN 24 is not shown? iv) What are the filled and unfilled boxes in figure 8a? v) similarly colored solid lines used for LCN 139, 11, 77, 76 vi) indistinguishable colors for various p values used in Fig 7a vii) error bars need be added in 4c, 4d

2) In all these unconnected glacial ponds, particularly those with significant glacier coverage in their basin, could it be checked if the corresponding glacier drains into the pond?

3) As acknowledged by the authors the study area is full of debris covered glaciers. The applicability of the glacier melt model used for debris covered glacier must be discussed.

4) It is known that SOI toposheets derived from winter time areal imagaries may contain significant errors. Some of the authors have published results using high resolution Corona KH4 images from 1962 in this area. Could the same images be used to verify the baseline 1962 extents of the ponds studied? Corona data should help in filliing the large time gap between 1962 nd 1992.

5) Which climate data is used for the correlation studies? Pyramid data or reanalysis/gridded products? If pyramid data is used then what is need of describing the others? If the gridded/reanalysis data are used then why not study the correlations for a period longer than the time-window of 2000-2013? What happens if the analysis is extended to all the ponds and for the duration of 1962-2013 using the GPCC precipitation data?

6) The details of the computation of the mean pond area change and its uncertainty may be explicitly pointed out.

7) While the authors do a good job of pointing the reader to the appropriate references, at times they may become distractions. For example while both the following cited references are great read in their own ritght, the citations here may be a bit far-fetched - "The current study is focused on the southern Koshi (KO) Basin, which is located in the eastern part of central Himalaya (CH) (Yao et al., 2012; Thakuri et al., 2014) (Fig. 2)". Also refer to Major comment (2) in this context.

8) How are the periods of 1992-2000, 2000-2008, 2008-2011 and 2011-2013 used in table 3 selected?

9) The conclusion has lengthy discussions about glacier changes and only a few words on the multi-temporal pond extent data described in the rest of the paper. The connection between the claimed signal from pond area change and glacier changes in the region is not explicitly mentioned as well.

10) Some typographical errors: l 67 "opeping" l 122 : "montly comulated" l 194: morphologicalal

---

## Referee Comment (RC3) · P. Buri (Referee) · 20 May 2016

Summary:

The authors investigate surface area changes of ponds over a period of fifty years (1963-2013) in a high-elevation Himalayan region using a topographic map (1963) as well as various Landsat satellite images (1992-2013). They relate the observed area changes to precipitation, temperature and glacier melt trends. The meteorological dataset used in this study is based both on a high-elevation weather station in the catchment (operating since the mid 1990's) and regional gridded and reanalysis data used to extend the record back in time to the 1960's, for which the authors have the first inventory of ponds (1963). The authors find a high sensitivity of ponds to a change in climate and try to use water bodies as proxies to detect behavior of precipitation and

glacier melt.

General comments:

The paper is generally well written and structured in a clear way. However, I have some major issues regarding the methods applied that question partly your conclusions. In addressing these points (mentioned below) the paper may could be improved considerably and your original dataset and conclusions could be presented in a concise way and more scientific value could be added to your work. You relate changes in the climate to changes in the lake areas, as meteorological parameters are often represented in a highly limited way in remote and high-elevation regions. This is an interesting but also novel concept and addresses a relevant scientific question within the scope of the journal, as e.g. temperature and precipitation build the base for many research questions in various fields of the cryosphere. However, it is questionable if the approach used in this study can be used to reconstruct changes in the climate as lakes respond to many inputs as say yourself, so pond area is only an integrated variable (see point 4 below). The provided references appropriate and referenced in a helpful way in the text. At least one new study (published after submission of this manuscript, see major point 1 below) should be added. The statistical analysis and the results, respectively, are not fully clear everywhere in the manuscript (e.g. Table 3, see point 3 below). The methods description is rather complete, with methods explained either directly in the text or by referring the reader to further literature. They major issues to address are listed here:

Major issues:

1) Satellite images used for the analysis:

First, you need to indicate in the main text, including abstract, which satellite images you use (not only in the supplement) as this is a key information. You use Landsat (from Table 2 of supplement) and there might be an issue of too coarse resolution with Landsat. Pond area strongly depends on the accuracy of the derived outlines. This is

a key issue and you should provide some errors in your delineation, mainly due to the resolution of the images. Watson et al. (2016), looking at supra-glacial ponds though, show that resolution is an issue and they state that Landsat products cannot be used for this purpose. So may cite this paper (which came out after your submission) and also consider that issue. Maybe your ponds are very big and not affected by the coarse resolution of Landsat? A clear advantage of Landsat is that it allows going back in time – what the higher resolution products cannot as they are all for recent years. Also, from Table 3 of supplement there is an ALOS image listed, although it is not clear what is that used to. ALOS has a different resolution and so this should be discussed.

2) Degree-day model for glacier melt:

The use of a degree-day model for glacier melt might be a key limitation, as this has been shown to be very sensitive to temperature fluctuations. Therefore the estimates of "glacier melt" might be erroneous, and responding too much to changes in temperature. I would suggest that you perform calculations with a better model. Also, a key concern is that you use a constant melt factor from another study - the model needs calibration. If you cannot do this, you should perform an uncertainty analysis by varying this factor in a given range. In addition, why did you only use one factor and not two for snow and ice? I would strongly recommend that you: 1. do an uncertainty analysis and see how sensitive your results are to changes in the degree-day factor 2. use a more appropriate model

3) Table 3:

There are some very contrasting changes and it is not entirely clear how these values were derived: e.g. for ponds with glacier coverage <5% from 1963 to 2011 there is a decrease of -7% (+-6%, which is a lot) and from 1963 to 2013 (only two years apart), there is a decrease of -25%. This could be due to accuracy in the delineation and the use of different data sources rather than real changes. Also, why are changes from intermediate periods, i.e. 2000 to 2013 (or 2000 to 2011), not shown in the table?

4) Aim of the paper:

You want to study lakes as proxies for climate, but you cannot indeed as lakes changes can only be explained if changes in a variety of climatic and glacier variables are known. What you can do is relating lake changes to climate and glacier changes and see if there is a consistent interpretation for both. This has to be changed in the intro and the paper in general.

5) Debris-covered and debris-free glaciers:

I strongly recommend that you carry out your analysis of glacier area changes separately for the two categories debris-covered and debris-free glaciers, and provide figures of how much of the glacier area in the catchment is covered by debris. Debris covered glaciers are known to shrink little in area and that area change is not a good indicator of glacier changes and melt (see e.g. lines 251-252).

Specific comments:

I think you should also analyze and discuss the fact that some ponds undergo geometrical changes over such a long time due to changing boundary conditions. Depending on the location and size of a water body, possibly enhanced or reduced sediment supply from glaciers, landslides etc. could change the lake area considerably. Also groundwater may play a role for the hydrology of some ponds. And if you think these processes are negligible, mention this in the text at the beginning in the introduction or at the end in the discussion. Regarding the topographical analysis, there are some hidden steps which need to be explained better in the text, e.g. selection of basins, aspects etc. (see specific comments below) or how you distinguish between a connected and an unconnected pond, i.e. how far the latter is located from the glacier tongue. There are sections in the text which need to be improved. Due to many different datasets, time periods and pond categories it is sometimes hard to follow step by step the selection and analysis of the data (is a certain result about ponds/season/years etc.). This could be improved by 1) using a clearer structure and repeating more frequently corresponding information in the text, and 2) splitting long sentences. This clarity is also lacking in a few figures, where it is sometimes not possible to get the right information of all plot elements. Some additional legend elements and a more precise caption would help substantially in these cases (see technical corrections below).

Technical corrections (text):

Line 11, '. . .ponds not directly connected to glaciers,', try to give a clearer definition to avoid mixing physical and hydrological connection, something like '. . .ponds not in direct contact with glacier ice' could fit.

Lines 14-15, wrong word order, write '. . .unconnected ponds have decreased significantly by approximately 10% over the last fifty years (1963-2013 period).'

Also: '10%' is area or number? Needs to be specified as it is ambiguous like that.

Line 16, word missing within 'We inferred an increase in precipitation occurred until. . .'

Line 22, 'remoteness' is another main reason.

Line 36, '. . . body of research. . .', try to use a better word.

Lines 46 and 54, '. . .high Asian mountains. . .', better to use 'high mountain Asia' or 'Asia's high mountains'.

Line 47, 'decreased evaporation', add explanation why evaporation was assumed to have decreased.

Lines 59-61, wrong word order, write 'Therefore the potential risk of GLOFs in the Himalaya has been,. . .'.

Line 61, '. . .these lakes', which type do you mean here?

Line 67, write '. . .opening'.

Line 69, '. . .only influenced by glacier melting and precipitation.', is this valid? What about e.g. evaporation, ground water, avalanches?

Line 70, write '. . .lakes to potential indicators. . .'.

Line 72, not sure you can use 'evapotranspiration' here, but also in several other parts of the text. Don't you mean 'evaporation' in general? Sometimes you use evaporation, sometimes evapotranspiration. Try to be consistent.

Line 73, write 'A valuable. . .'.

Line 79, it seems to me that Hamerlik et al. (2013) used a threshold of 1 ha (page 3), better cite Biggs et al. (2005).

Line 94, '. . .characterized by. . .', be more concise.

Line 97, 'For the last twenty years. . .', give specific years.

Lines 97-98, wrong word order.

Line 106, '. . .these glaciers. . .', which glaciers?

Line 118, write '. . .and subsequently expanded continuously. . .'.

Line 122, write '. . .monthly cumulated. . .'.

Lines 125 and 127, write 'Jensen-Haise model'.

Lines 136, gap between '. . .Unit-Time. . .'.

Lines 138, gap between '. . .Prediction-Climate. . .'.

Line 154, write '. . .through. . .'.

Lines 156-159, sentences about selection are confusing, try to explain this more clearly.

Line 172, specify why you selected this T-index model. See also major comments above.

Line 174, '. . .close to the SNP.', explain better why this field study on Glacier AX010

is the best solution and suitable in your opinion, specify where this glacier is located, which region, climate etc. See also major comments above.

Line 175, why didn't you apply the daily temperature per elevation band of each glacier?

Line 178, delete 'Such'.

Line 179, write '. . .through. . .'.

Line 180, use proper reference instead of URL-address.

Line 182, use proper reference instead of URL-address.

Line 185, maybe more correct to use 'mountainous terrain' or 'steep terrain'.

Line 189, use proper reference instead of URL-address.

Line 190, write '. . .effects as decribed in Salerno. . .'.

Line 194, write '. . .morphological. . .'.

Line 205, add reference to '. . .in the software R. . .'.

Line 213, '. . .trends has been tested. . .' on how many years? Isn't there a minimum of years to be able to speak about trends?

Line 233, description for Figure SI2b confusing and not consistent with actual plot.

Line 240, remove 'very' or use 'relatively'.

Line 240, write '. . .oriented towards south-southeast. . .'.

Lines 243-245, wrong word order, write '. . .in the last fifty years (1963-2013).'.

Also: 10% is ambiguous: is this area or number?

Line 257-258, This depends on the status of the glaciers, see e.g. Pellicciotti et al., 2010. You can have a decrease in area and decrease in glacier melt.

[Figure]

Lines 258-259, avoid using two times 'However...'.

Line 261, '...extremely broad...' not clear to me what you mean here, use clearer/better word(s).

Line 284, replace 'These authors...' with 'They...'.

Lines 284-287, wrong word order, write 'They observed...'. Too long sentence, make two out of it.

Line 291, delete 'both'.

Line 296, write '...than the mean...'.

Line 298, write '... more than the...'.

Line 303, what do you mean with '...relevant...'? Try to be more clearly. Also: mentioning 'maximum monsoon temperature' and 'glacier melt' as main drivers of change is somehow redundant in my opinion, as the last is clearly directly dependent of the former one in your calculations. Maybe explain here better the dependencies.

Lines 303-305, too long and complicated sentence, untangle and make two out of it.

Line 315, write '...basin...'.

Line 317, maybe you can mention, that based on your findings it can be clearly seen, that glaciers act as buffers of the hydrological cycle.

Line 328, remove 'very' or use 'relatively'.

Line 330, write 'compare'.

Lines 333-335, wrong word order and too long sentence. Write 'The surface area of ponds-without glaciers strongly decreased (-25±6%, p<0.001) from 1963 to 2013. In contrast, the surface area of ponds-with-glaciers decreased much less (-6±2%, p<0.05) for the same period.'

Also: refer to Table 3 in that sentence.

Lines 361-362, contradiction to line 355 and Figure 9b., should be the other way round I suppose.

Lines 362-363, here you could think about glacier morphology to further explain differences in glacier melt at different elevations (area, steepness, debris), if this is valid in your case study.

Line 369, be more precise when using the term 'glacial ponds' in order to separate them from supraglacial ponds etc.

Line 372, missing word(s) in 'The continued shrinkage of glaciers likely due to. . .'.

Line 376, avoid using 'study' two times.

Line 377, I wonder if the behavior of precipitation and glacier melt can be detected separately based on tracked pond areas. Maybe you can state something about this here.

Lines 382-387 & lines 389-391, did you directly observe constant (until the 1990s) or reduced glacier melt (in the early 2000s) or is this assumption based on the decreased max. air temperatures? It would be good if you could add here more background from your findings.

Line 403, write '. . .other climatic. . .'.

Line 409, verb missing.

Technical corrections (tables/figures):

Table 2:

Line 629, write '. . .of all considered. . .'.

Pond area, rounding error for max. value in 2nd and 3rd column (56.3 vs. 56.2)?

Basin, maybe you can add once in the paper how the basin is defined (='hydrological' catchment?) and how you calculated it (algorithm?).

Basin aspect, did you consider the calculation for directional values? Mean, median, range etc. of aspects have to be derived carefully, as e.g. the mean and median of the three values $45°$, $345°$ and $360°$ doesn't make sense if calculated normally. Add a short note how you deal with this once in the paper where 'aspect' occurs first.

Also: How did you derive the mean basin aspect? Add used method ('vectorial mean').

Glacier aspect, same as 'basin aspect', see comment above. Here it seems that the median is not within the range.

Table 3:

Asterisks, what do they stand for? Statisitcal significance level? Add explanation.

Table 4:

Basin aspect, again, how did you coalculate mean and median basin aspect(s)? Asterisks, what do they stand for? Add explanation.

Figure 1:

Line 684, you could add the source of the two pictures.

Figure 2:

a), use decimal degrees as written in text (line 91).

Also: black triangle and 'SNP' somehow misleading in inset map.

b), write '...isotherms corresponding...'.

Also: write 'max. temperature'

Line 715, remove '...'.

Figure 4:

Low image quality, especially axis labels. Try to improve.

Also: change x-axis labels to more 'intuitive' years, e.g. 1980, 1985,... and add year labels to all subplots a-d for better readability.

b), write 'Precipitation (anomaly)'

Figure 6:

Low quality, labels and lines.

Also: units missing.

a), y-range seems to be too small, missing points.

Also: wrong labels both at y-axis and in legend ('cumulate').

b), the left and right y-axes seem to be shifted vertically.

Line 777, a) and b) mixed?

Line 779, write '...Figures...'.

Figure 7:

Especially subplots a) and c) too small.

Also: size of circles in subplots b) and d) not clear, explanation below not clear as well.

Line 783, write 'Increased pond surface areas' and 'Decreased pond surface areas'.

Lines 785-786, description of subplots a) and c) not consistent with actual titles in plot (with/without glaciers).

Figure 8:

Add units for right y-axes (precipitation, melt). Also: make lines and bars in both subplots identifyable, label them.

Figure 9:

Low quality, too small (axes labels).

Technical corrections (supporting information):

Figure SI1:

Last sentence in caption: write 'In Table 1 the relevant coefficients of correlation are reported.'.

Figure SI2:

a), add more space in between x-axis-labels. b), change x-axis-labels to more 'intuitive' years (e.g. 1980, 1985, . . .).

Figure SI3:

Very low quality of all labels, axes, wrong number of digits etc., too small. Also: add units or write that the anomalies are relative or dimensionless.

Figure SI4:

Low quality of all labels, too small. Second last sentence in caption: write '. . .considering Tmax and Tmean.'.

References:

Biggs, J., P. Williams, M. Whitfield, P. Nicolet and A. Weatherby, 2005. 15 years of pond assessment in Britain: results and lessons learned from the work of Pond Conservation. Aquatic Conservation: Marine and Freshwater Ecosystems 15: 693–714.

Pellicciotti, F., A. Bauder and M. Parola. Effect of glaciers on streamflow trends in the Swiss Alps. Water Resources Research, 46: W10522.

Watson, C.S., D.J. Quincey, J.L. Carrivick and M.W. Smith, 2016. The dynamics of

supraglacial water storage in the Everest region, central Himalaya. Global and Planetary Change 142: 14–27.

---

## Author Comment (AC3) · 10 Jun 2016

**Banerjee (Referee)**

This paper reports on the surface area changes of unconnected glacial ponds in the south of Mt Everest during the period 1962 to 2013 using maps and satellite images. This time-series data is analysed to identify the drivers of the change using statistical analysis of the correlations with available meteorological data. However, the present draft may greatly benefit from a more careful analysis of this very interesting data set, and also a slightly more systematic description of the methodological details.

**Comment:** we thanks the reviewer for the revision. Generally, we hope to have suitably followed the suggestion received in particular in relation to the new analysis and the more detailed methodology.

My major concerns are as follows:

1) While it has been argued at the outset rather briefly that lakes and ponds are sensitive indicators of climate change, this point demands more serious consideration. The cited references of Beniston et al, 2006 do not seem to discuss lake/pond, while the other referred article by Burasachi et al, 2005 does not include a relevant discussion of climate sensitivity of the lake/pond area and particularly of the response time scales.

**Answer:** Thanks for the suggestion. The references are wrong as suggested by the reviewer.

**Correction: the right reference is Smol and Douglas, 2007. PNAS**

The temporal variation of the surface area of a given pond must be controlled by 1) the balance between water in and out - therefore by the climate, and 2) the bathymetry. But, any attempt to infer climate signal from sparse point measurements of such a time series has to take into account the relevant time-scales associated with response to the fluctuating climate variables.

**Answer:** we hope having followed all revision provided by reviewers, the suggestion provided by the reviewed could be suitably addressed.

For example in figure 3, some of the biggest ponds/lakes (eg LCN77, LCN24) show large (_5n%) increase/decrease in their area in a month's time, indicating a strong control of high frequency changes of the climate variables. For the rest of the ponds which are even smaller in size, these high frequency noise would presumably be even larger. How can this sparse time series with high frequency 'noise' that is of similar magnitude as the low frequency signal (_10n% change over 50 years), possibly be used to infer low frequency changes of the climate? In this context it may be noted that glacier length fluctuations can be inverted for temperature change as their slow response makes them immune to high frequency noise.

**Answer:** Figure 3 shows that some single pond presents singularly an Oct-Dec dispersion of around 5%. However, the same figure points out that just averaging this information on a population only a little bit larger, the dispersion between October and December becomes almost zero (1%). Climatic inferences from the behavior of ponds population surely needs to consider the widest number of ponds as possible in order to reduce the dispersion due to the local conditions of each lake. The same approach is used also in dendrochronology where a lot of cores are sampled and analysed.

**Correction: the suggestion of the reviewer has been considered carefully inserting these concepts in the new text.**

Similar large fluctuations are also seen in the annual rates reported in Table 3: During 1992 to 2011, rates are very small or insignificant and then there is a very large (1 to 2 order of magnitude larger than the background) spike during 2011-2013. In fact this spike dominates the mean. Is this a signal from a particular short-lived event picked up due to sparse sampling or a real climate change signal? Surprisingly, no such sharp changes are seen in the precipitation or glacier melt data during 2011-2013 as presented here. This needs to be considered very very carefully before accepting the interpretation offered by the authors here. Further, this issue of high frequency noise can not be overcome simply by averaging over a large set of ponds from the same region, as they are all seeing a strongly correlated noise due to their spatial proximity. And of course, practical limitations like unavailability of suitable images etc would prevent a higher temporal resolution.

> **Answer:** In relation to the abrupt change observed by the reviewer (-7% vs -25%,i.e., -18%), we can start observing Table SI2. The resolution of the two images is the same. Moreover giving a look at fig. 8 Fig. 8. Probably here it looks much less strange. From 1992 to 2011 the decreasing is 20% (the computation can be done also from the table 3 from +13% to -7%). Surely -18% in two years is a lot, but in in line with the decreasing of precipitation observed since the early '90s (Fig. 8). Furthermore the behavior of surface are change has been observed significantly correlated with precipitation.
>
> **Correction: this concept has been discussed in the text.**

2) While the authors have employed a careful statistical methods to derive their conclusion regarding the climate signal, some simple physical considerations might strengthen their analysis. For example, the climate data (reanalysis/gridded) used is from the grid point that is closest to the Pyramid station. Would not be better to use the grid point closest to a given pond for analysing the area change data for that particular pond? This choice might have led to serious biases in the results as pointed out below.

> **Answer:** Unfortunately, the grid resolution for Era Interim and GPCC does not allow to use the grid point closest to a given pond, because this point is common to all points and at same time it is the same grid node used in the comparison with the land wheatear station. We agree with the reviewer. In fact in the introduction we wrote: "their use for climate change impact studies at the synoptic scale must be performed with caution due to the absence of weather stations across the overall region, which limits the ability to perform land-based evaluations of these products". The added value of this work to have carried out a land-based evaluation of these products. Probably, the comparison presented in this work in the unique case where this comparison has been done for a so long period of time in the overall Himalayan range due to other long time climatic series do not exist in the region at so high elevation.
>
> Moreover the comparison between ponds surface area and climatic variables is done with Pyramid data. To avoid further misunderstanding we tried to clarified these concept as specified below.
>
> **Correction: the method section has been rewritten to clarify the methodological approach followed in the paper: Moreover a map of Nepal showing the location of all 64 considered ponds and the grid/reanalysis nodes has been inserted in the Supplementary Material. In this way it is clear the comparison between the resolution of grid/reanalysis products and the distribution of the 64 considered lakes.**

All the 'ponds with glaciers' (LCN 24,9,3,68) that show significant correlation with glacier-melt, are located in the Khumbu valley, within may be five km of the Pyramid station. So, how can one be sure about the controlling factor behind this pattern - Is it the glacier cover as claimed, or it is just the proximity of the grid point? In fact, data from LCN11 in the same valley has a relatively large correlation coefficient (_0.5, though

probably not significant) with calculated glacier melt, while far-away 'ponds with glaciers' (LCN 76 and 77) has small (0.2-0.3) correlation with the glacier melt. This requires explanation.

**Answer:** please see the answer above.

Incidentally, there seem to be some ambiguity regarding the definition of two pond classes: with and without glaciers. Table 3 uses 5n% as a threshold; text gives a threshold of 10n%; Table 4 says LCN3 has 30n% glacier cover, while Figure 3 claims LCN3 is a pond without glacier. These differences need to be clarified and the sensitivity of the conclusions to this choice of threshold value may be discussed.

**Answer:** the threshold of 5% reported in the heading of Table is a mistake. The second suggestion is not clear when the reviewer says: "while Figure 3 claims LCN3 is a pond without glacier"… we do not know where this is discussed.

**Correction:** Table 3 has been corrected.

Also, the authors may discuss the spatial pattern of changes as seen in figure 7. For example, looking at this figure it seems statistically significant differences may emerge in trends from the set of ponds near Ngzumpa glacier (Gokyo valley ) and Khumbu glacier (Khumbu valley), irrespective of glacier cover extent. If so, then what is the relevant control, having more than 5n% glacier area or the ponds being in the same valley?

**Answer:** According to the suggestion  we tested the significance of possible differences of surface area changes (1992-2013 period) among ponds located in different river basins. Moreover we tested the significance of possible differences of surface area changes among ponds with different glacier cover.

**Correction: a new figure in the Supplementary Material has been added showing box-plots representing  surface area changes of ponds located in different river basins. A parametric test (ANOVA) shows no significant difference among the different river basins. The same Figure reports box-plots representing surface area changes among ponds with different glacier cover. A parametric test (ANOVA) shows in this case significant differences between ponds with different glacier cover within basins.**

In addition, the ponds with glacier cover seem to be larger (table 2). Could it be that the difference in shrinkage are correlated with pond size? A possibly larger intrinsic climate sensitivity of the smaller ponds could be an alternate explanation for the differences seen between the signal from the two class. This possibility needs to be ruled out as well to justify the conclusions reached.

**Answer:** According to the suggestion  we tested the significance of possible differences of surface area changes (1992-2013 period) among ponds with different size.

**Correction: a new figure in the Supplementary Material has been added showing box-plots representing  surface area changes among ponds with different size. A parametric test (ANOVA) shows no significant difference.**

Other comments:

1) Many of figures presented needs to be carefully redone, checking the axes labels for missing units, choosing proper x and y range so that all data-points are seen, putting legends that are missing, giving complete and accurate plot captions etc. Some examples: i) what are the units of vertical scales in figure 3, 4a, 4b, 6a, ... ii) Figure 3 horizontal axis: tics read 06,07,07,08,... . Also horizontal separation of the points

are inconsistent with time stamps given in table SI4. iii) what are the criteria for the selecting the ponds whose records are presented in figure 6? why LCN 24 is not shown? iv) What are the filled and unfilled boxes in figure 8a? v) similarly colored solid lines used for LCN 139, 11, 77, 76 vi) indistinguishable colors for various p values used in Fig 7a vii) error bars need be added in 4c, 4d

> **Answer:** Following the comments received by all the reviewers all figures and tables have been checked and redone.
>
> **Correction: i) the units have been written in the caption; ii) checked and redone; iii) there was an error, LCN24 has been inserted; iv) the legend has been added; v) corrected; vi) corrected; vii) error bars have been added**

2) In all these unconnected glacial ponds, particularly those with significant glacier coverage in their basin, could it be checked if the corresponding glacier drains into the pond?

> **Answer:** The hydrological basins have been digitalized using ArcGIS® hydrology tools as carried out by other authors (e.g., Pathak et al., 2013), each basin has been then visually checked.
>
> **Correction: this methodological aspect has been inserted.**

3) As acknowledged by the authors the study area is full of debris covered glaciers. The applicability of the glacier melt model used for debris covered glacier must be discussed.

> **Answer:** The glaciers within the pond basins are not debris covered. In this region debris covered glaciers are usually glaciers of a certain size with a developed flat ablation area. In all considered pond basins, the glacier are very small, steep (31°), clings to the mountain peaks, without having developed debris covered ablation area.
>
> **Correction: following the suggestion of the reviewer we specified in the text these features of glaciers within the pond basins.**

4) It is known that SOI toposheets derived from winter time areal imagaries may contain significant errors. Some of the authors have published results using high resolution Corona KH4 images from 1962 in this area. Could the same images be used to verify the baseline 1962 extents of the ponds studied? Corona data should help in filliing the large time gap between 1962 and 1992.

> **Answer:** We did not used only Corona image for digitalizing all 64 considered ponds because many of them in this image are snow-covered, but, we checked the quality of the map comparing the size of some ponds in both data sources.
>
> **Correction: "The topographic map of the Indian survey of 1963 (hereafter TISmap-63, scale 1:50,000) was used to complement the results achieved using the declassified Corona KH-4 (15 Dec 1962, spatial resolution 8 m). Thakuri et al., 2014 describe the co-registration and rectification procedures applied to the Corona KH-4 imagery. Unfortunately, on these satellite images many ponds are snow-covered. Therefore here the ponds surface area digitalized on TISmap-63. The accuracy of this map has been tested comparing the surface areas of 13 ponds digitalized on both data sources (favoring the cloud and shadow free ponds). Figure SI1 shows the proper correspondence of these comparisons. Furthermore, in order to estimate the mean bias associated with TISmap-63, we calculated the mean absolute error (MAE) (Willmott and Matsuura, 2005) between data, which resulted sufficiently low (3.6%), assuring in this way the accuracy of ponds surface area digitalized on TISmap-63."**

5) Which climate data is used for the correlation studies? Pyramid data or reanalysis/ gridded products? If pyramid data is used then what is need of describing the others? If the gridded/reanalysis data are used then why not study the correlations for a period longer than the time-window of 2000-2013? What happens if the analysis is extended to all the ponds and for the duration of 1962-2013 using the GPCC precipitation data?

**Answer:** As described above the correlation studies have been done using the Pyramid data due to the continuous series of annual ponds surface area are available only for the 2000-2013 period and land meteorological data are available for 1994-2013 period. This explains, answering to the reviewer, why a time-window longer than 2000-2013 does not exist. Extending the analysis to the 1962-2013 is not possible because before 2000 we have just two years in which it was possible to digitalize the ponds (1963 and 1992).

Gridded/reanalysis data are not used here for correlation studies, but to obtain information, as written in the paper, on climatic trends in the antecedent period (before 1994). For this reason they have been compared with land data and the best products have been chosen.

Correction: following this suggestion and all other comments received in particular from reviewer 1 a need of clarification clearly emerges. Therefore the method section of the paper has been restructured and these concepts clarified.

6) The details of the computation of the mean pond area change and its uncertainty may be explicitly pointed out.

**Answer:** as even suggested by another reviewer the section related the uncertainty computation is too hermetic.

**Correction: Therefore it has been rewritten.**

7) While the authors do a good job of pointing the reader to the appropriate references, at times they may become distractions. For example while both the following cited references are great read in their own ritght, the citations here may be a bit far-fetched - "The current study is focused on the southern Koshi (KO) Basin, which is located in the eastern part of central Himalaya (CH) (Yao et al., 2012; Thakuri et al., 2014) (Fig. 2)". Also refer to Major comment (2) in this context.

**Correction: we deleted Yao et al., 2012**

8) How are the periods of 1992-2000, 2000-2008, 2008-2011 and 2011-2013 used in table 3 selected?

**Answer:** the periods have been selected in relation to the availability of satellite imagery

9) The conclusion has lengthy discussions about glacier changes and only a few words on the multi-temporal pond extent data described in the rest of the paper. The connection between the claimed signal from pond area change and glacier changes in the region is not explicitly mentioned as well.

**Answer:** the conclusions have been rewritten and the "connection" has been more explicitly mentioned

10) Some typographical errors: l 67 "opeping" l 122 : "montly comulated" l 194: morphologicalal
**Answer:** The suggestion has been followed

---

## Author Comment (AC4) · 10 Jun 2016

**P. Buri (Referee)**

Summary:

The authors investigate surface area changes of ponds over a period of fifty years (1963-2013) in a high-elevation Himalayan region using a topographic map (1963) as well as various Landsat satellite images (1992-2013). They relate the observed area changes to precipitation, temperature and glacier melt trends. The meteorological dataset used in this study is based both on a high-elevation weather station in the catchment (operating since the mid 1990's) and regional gridded and reanalysis data used to extend the record back in time to the 1960's, for which the authors have the first inventory of ponds (1963). The authors find a high sensitivity of ponds to a change in climate and try to use water bodies as proxies to detect behavior of precipitation and glacier melt.

General comments:

The paper is generally well written and structured in a clear way. However, I have some major issues regarding the methods applied that question partly your conclusions. In addressing these points (mentioned below) the paper may could be improved considerably and your original dataset and conclusions could be presented in a concise way and more scientific value could be added to your work. You relate changes in the climate to changes in the lake areas, as meteorological parameters are often represented in a highly limited way in remote and high-elevation regions. This is an interesting but also novel concept and addresses a relevant scientific question within the scope of the journal, as e.g. temperature and precipitation build the base for many research questions in various fields of the cryosphere. However, it is questionable if the approach used in this study can be used to reconstruct changes in the climate as lakes respond to many inputs as say yourself, so pond area is only an integrated variable (see point 4 below). The provided references appropriate and referenced in a helpful way in the text. At least one new study (published after submission of this manuscript, see major point 1 below) should be added. The statistical analysis and the results, respectively, are not fully clear everywhere in the manuscript (e.g. Table 3, see point 3 below). The methods description is rather complete, with methods explained either directly in the text or by referring the reader to further literature. They major issues to address are listed here:

> **Comment:** we thanks the reviewer for the revision of the paper. Generally, we hope that in the new version the key messages could emerge more clearly. All the suggestions have been followed. A new overall methodological section have been introduced.

Major issues:

1) Satellite images used for the analysis:
First, you need to indicate in the main text, including abstract, which satellite images you use (not only in the supplement) as this is a key information. You use Landsat (from Table 2 of supplement) and there might be an issue of too coarse resolution with Landsat. Pond area strongly depends on the accuracy of the derived outlines. This is a key issue and you should provide some errors in your delineation, mainly due to the resolution of the images. Watson et al. (2016), looking at supra-glacial ponds though, show that resolution is an issue and they state that Landsat products cannot be used for this purpose. So may cite this paper (which

came out after your submission) and also consider that issue. Maybe your ponds are very big and not affected by the coarse resolution of Landsat? A clear advantage of Landsat is that it allows going back in time – what the higher resolution products cannot as they are all for recent years. Also, from Table 3 of supplement there is an ALOS image listed, although it is not clear what is that used to. ALOS has a different resolution and so this should be discussed.

> **Answer:** As suggested by the reviewer the supraglacial lakes in Mt Everest Region are very small. According to Watson et al. (2016) their size range from 0.09 to 0.36 $10^4$ m$^2$, while the unconnected ponds in the same region (this study) are on average 1.1 $10^4$ m$^2$, i.e., an order of magnitude larger. This is not the unique difference between the two kind of ponds. As described in the text, supraglacial ponds are strictly connected with glacier dynamics, thus, as describe by many authors (and by the same Watson et al. (2016)) their measurement is very uncertain. Landsat imagery is surely too coarse for these ponds.
>
> Considering unconnected ponds, in general, we tracked the pond surface changes in many papers (Tartari et al., 2008, Thakuri et al., 2015; Salerno et al., 2012, Salerno et al., 2014). We wrote a specific work (Salerno et al., 2012), on the uncertainty related to the measurements of lakes from satellite imagery in the region, which is referenced also by Watson et al. (2016). In the methodological section there is a section devoted to the uncertainty of measurements.
>
> Table 3 is a general summary of surface area changes related to all 64 considered ponds glaciers located within the basins. In the previous version of the paper was not explicitly written that the same table reported the uncertainty of measurements. This could have confused the reviewer, which though that we did not consider and discuss the uncertainty of measurements.
>
> The ALOS was used to track the pond surface areas in 2008, this image was preferred considering the better resolution. In fact in table these period presents uncertainties slightly lower.
>
> **Correction: 1) the methodological section related to the uncertainty of measurements has been extended. 2) we corrected the caption of Table 3. Along the paper, where it was omitted, the uncertainty has been associated with relevant difference of measurement. The satellite images used for the analysis have been also reported in the main text and in the abstract.**

2) Degree-day model for glacier melt:
The use of a degree-day model for glacier melt might be a key limitation, as this has been shown to be very sensitive to temperature fluctuations. Therefore the estimates of "glacier melt" might be erroneous, and responding too much to changes in temperature. I would suggest that you perform calculations with a better model. Also, a key concern is that you use a constant melt factor from another study - the model needs calibration. If you cannot do this, you should perform an uncertainty analysis by varying this factor in a given range. In addition, why did you only use one factor and not two for snow and ice? I would strongly recommend that you: 1. do an uncertainty analysis and see how sensitive your results are to changes in the degree-day factor 2. use a more appropriate model

> **Answer:** This paper does not aim to provide an accurate estimation of the magnitude of the melt released from glaciers located in the pond basins. In fact, its value has never been discussed and mentioned. The melt factor could be unsuitable, but if it was wrong no analysis would be compromised. We compared its 2000-2013 trend vs the pond surface areas, and the correlation analysis is independent from the magnitude of the compared series. Consequently, we do not need different factors for snow and ice and to make a sensitive analysis.

Being interested in the melt trend and not in its absolute magnitude and considering that these small glaciers are ungauged, we do not more sophisticated melt models, which consider specific geometries and differentiated melt factors. We are aware of the autocorrelation between the maximum temperature and glaciers melt calculated from this variables, i.e., their fluctuation are similar. The added value is only due to that the positive temperature calculated for each glacier (elevation bends) are able to generate a melt, which we found to be significant related to the observed pond surface area changes. If ponds (and glaciers) were located some hundred of meters at higher elevation, surely the melt and Tmax would be less correlated and the application of the degree-day model would look less trivial. What is the knowledge contribution of the application of the degree-day model in this contest? Maximum temperature trend is here demonstrated to be responsible of processes able to modify the pond surface area. How processes? Glacier melt is a reasonable factor, due to we find significant relationships when glaciers are present in the pond basins, and no relationship with Tmax when glacier are not present in the basin.

**Correction: these concepts has been inserted in the text.**

3) Table 3:

There are some very contrasting changes and it is not entirely clear how these values were derived: e.g. for ponds with glacier coverage <5% from 1963 to 2011 there is a decrease of -7% (+-6%, which is a lot) and from 1963 to 2013 (only two years apart), there is a decrease of -25%. This could be due to accuracy in the delineation and the use of different data sources rather than real changes. Also, why are changes from intermediate periods, i.e. 2000 to 2013 (or 2000 to 2011), not shown in the table?

**Answer:** In the right of the Table 3 changes for each intermediate period are all referred to 1963, because they are expressed as commutative loss. Having fixed the reference year this kind showing results allows to create a trend. In fact these data are the same used in Figure 8. I f we were interested in the acceleration for each period, the same Table on the left provides the relative annual rate (for each period in this case). These data are discussed in Table 7. So you can directly compare periods.

In relation to the abrupt change observed by the reviewer (-7% vs -25%,i.e., -18%), we can start observing Table SI2. The resolution of the two images is the same. Moreover giving a look at fig. 8 Fig. 8. Probably here it looks much less strange. From 1992 to 2011 the decreasing is 20% (the computation can be done also from the table 3 from +13% to -7%). Surely -18% in two years is a lot, but in in line with the decreasing of precipitation observed since the early '90s (Fig. 8). Furthermore the behavior of surface are change has been observed significantly correlated with precipitation.

**Correction: the caption of the table has been changed to better clarify its content.**

4) Aim of the paper:

You want to study lakes as proxies for climate, but you cannot indeed as lakes changes can only be explained if changes in a variety of climatic and glacier variables are known. What you can do is relating lake changes to climate and glacier changes and see if there is a consistent interpretation for both. This has to be changed in the intro and the paper in general.

**Correction: the specific aims of the paper have been added.**

5) Debris-covered and debris-free glaciers:

I strongly recommend that you carry out your analysis of glacier area changes separately for the two categories debris-covered and debris-free glaciers, and provide figures of how much of the glacier area in the

catchment is covered by debris. Debris covered glaciers are known to shrink little in area and that area change is not a good indicator of glacier changes and melt (see e.g. lines 251-252).

**Answer:** The glaciers within the pond basins are not debris covered. In this region debris covered glaciers are usually glaciers of a certain size with a developed flat ablation area. In all considered pond basins, the glacier are very small, steep (31°), clings to the mountain peaks, without having developed debris covered ablation area.

**Correction: following the suggestion of the reviewer we specified in the text these features of glaciers within the considered pond basins.**

Specific comments:

I think you should also analyze and discuss the fact that some ponds undergo geometrical changes over such a long time due to changing boundary conditions. **A)** Depending on the location and size of a water body, possibly enhanced or reduced sediment supply from glaciers, landslides etc. could change the lake area considerably. Also groundwater may play a role for the hydrology of some ponds. And if you think these processes are negligible, mention this in the text at the beginning in the introduction or at the end in the discussion. **B)** Regarding the topographical analysis, there are some hidden steps which need to be explained better in the text, e.g. selection of basins, aspects etc. (see specific comments below) or how you distinguish between a connected and an unconnected pond, i.e. how far the latter is located from the glacier tongue. There are sections in the text which need to be improved. **C)** Due to many different datasets, time periods and pond categories it is sometimes hard to follow step by step the selection and analysis of the data (is a certain result about ponds/season/years etc.). This could be improved by 1) using a clearer structure and repeating more frequently corresponding information in the text, and 2) splitting long sentences. **D)** This clarity is also lacking in a few figures, where it is sometimes not possible to get the right information of all plot elements. Some additional legend elements and a more precise caption would help substantially in these cases (see technical corrections below).

**Answer:** A) the variably connected with "secondary" boundary conditions has been discussed in the conclusions; B) following the suggestions provided by the reviewer, accepting the specific comments provided below, we hope to have provided more details on these aspects; C) following the suggestion received by another reviewer, a section related to the overall mythology has been inserted; D) All figures and captions have been improved following the suggestions received by reviewers

Technical corrections (text):

Line 11, ': : :ponds not directly connected to glaciers,', try to give a clearer definition to avoid mixing physical and hydrological connection, something like ': : :ponds not in direct contact with glacier ice' could fit.

**Answer: done**

Lines 14-15, wrong word order, write ': : :unconnected ponds have decreased significantly by approximately 10% over the last fifty years (1963-2013 period).'

**Answer: done**

Also: '10%' is area or number? Needs to be specified as it is ambiguous like that.

**Answer: done**

Line 16, word missing within 'We inferred an increase in precipitation occurred until: : :'

    **Answer: done**

Line 22, 'remoteness' is another main reason.

    **Answer: done**

Line 36, ': : : body of research: : :', try to use a better word.

    **Answer: done**

Lines 46 and 54, ': : :high Asian mountains: : :', better to use 'high mountain Asia' or 'Asia's high mountains'.

    **Answer: done**

Line 47, 'decreased evaporation', add explanation why evaporation was assumed to have decreased.

    **Answer: done**

Lines 59-61, wrong word order, write 'Therefore the potential risk of GLOFs in the Himalaya has been,: : :'.

    **Answer: done**

Line 61, ': : :these lakes', which type do you mean here?

    **Answer: done**

Line 67, write ': : :opening'.

    **Answer: done**

Line 69, ': : :only influenced by glacier melting and precipitation.', is this valid? What about e.g. evaporation, ground water, avalanches?

    **Answer:** the mains terms of the water balance we consider at annual scale are as input, precipitation and glacier melt, and as output, the evaporation. If we considered ground water, avalanches we should also consider other terms as runoff, infiltration, seepage, sublimation…but this level of detail is not the aim of the work, and it is impossible to discern in this remote environments. These lakes are ungauged, remote. No information regarding the groundwater  is available at those elevations, avalanches are never computed in the water balances because are they are episodic not easily quantifiable events.

    Following the approach of other authors (e.g.,  Song et al., 2014; Wang et al., 2015, Salerno et al., 2015), precipitation, glacier melting, and evaporation are the main contributions in high elevated lake basins able to  explain the causes of lake changes

    **Correction: the approach followed by these authors has been inserted.**

Line 70, write ': : :lakes to potential indicators: : :'.

    **Answer: done**

Line 72, not sure you can use 'evapotranspiration' here, but also in several other parts of the text. Don't you mean 'evaporation' in general? Sometimes you use evaporation, sometimes evapotranspiration. Try to be consistent.

    **Answer: done**

Line 73, write 'A valuable: : :'.

    **Answer: done**

Line 79, it seems to me that Hamerlik et al. (2013) used a threshold of 1 ha (page 3), better cite Biggs et al. (2005).

    **Answer:** He initially used a threshold of 1 ha, but his analysis shown that the threshold was 2 ha (abstract)

Line 94, ': : :characterized by: : :', be more concise.

**Answer: done**

Line 97, 'For the last twenty years: : :', give specific years.

**Answer: done**

Lines 97-98, wrong word order.

**Answer: done**

Line 106, ': : :these glaciers: : :', which glaciers?

**Answer: done**

Line 118, write ': : :and subsequently expanded continuously: : :'.

**Answer: done**

Line 122, write ': : :monthly cumulated: : :'.

**Answer: done**

Lines 125 and 127, write 'Jensen-Haise model'.

**Answer: done**

Lines 136, gap between ': : :Unit-Time: : :'.

**Answer: done**

Lines 138, gap between ': : :Prediction-Climate: : :'.

**Answer: done**

Line 154, write ': : :through: : :'.

**Answer: done**

Lines 156-159, sentences about selection are confusing, try to explain this more
clearly.

**Answer: done**

Line 172, specify why you selected this T-index model. See also major comments
above.

**Answer:** the choice has been described above.

**Correction: this concept has been inserted in the text.**

Line 174, ': : :close to the SNP.', explain better why this field study on Glacier AX010
is the best solution and suitable in your opinion, specify where this glacier is located,
which region, climate etc. See also major comments above.

**Answer:** this glacier is a small debris free glacier, located in the Dudh Koshi valley in same climatic and geographic setting of glaciers studied in this paper, just outside the SNP in the southwest part (27°42'N, 86°34'E). Several studies exist on this glacier . It is a reference glacier for long monitoring of mass balance changes. Some papers: http://onlinelibrary.wiley.com/doi/10.1029/2005JD005894/full, www.pnas.org/content/108/34/14011.full.pdf

**Correction: this concept has been inserted in the text.**

Line 175, why didn't you apply the daily temperature per elevation band of each glacier?

**Answer:** the previous version was too hermetic and not clear.

**Correction: the text has been corrected according to the suggestion of specifying better the use of the elevation bands.**

Line 178, delete 'Such'.

**Answer: done**

Line 179, write ': : :through: : :'.

**Answer: done**

Line 180, use proper reference instead of URL-address.

**Answer: done**

Line 182, use proper reference instead of URL-address.

**Answer: done**

Line 185, maybe more correct to use 'mountainous terrain' or 'steep terrain'.

**Answer: done**

Line 189, use proper reference instead of URL-address.

**Answer: done**

Line 190, write ': : :effects as decribed in Salerno: : :'.

**Answer: done**

Line 194, write ': : :morphological: : :'.

**Answer: done**

Line 205, add reference to ': : :in the software R: : :'.

**Answer: done**

Line 213, ': : :trends has been tested: : :' on how many years? Isn't there a minimum of years to be able to speak about trends?

**Answer:** No there is not a minimum of years. However when a series is considered not such long, the associated significance should be considered with caution.

**Correction: This specification has been inserted in the text**

Line 233, description for Figure SI2b confusing and not consistent with actual plot.

**Answer: not done**. We did not understand the comment.

Line 240, remove 'very' or use 'relatively'.

**Answer: done**

Line 240, write ': : :oriented towards south-southeast: : :'.

**Answer: done**

Lines 243-245, wrong word order, write ': : :in the last fifty years (1963-2013).'.

Also: 10% is ambiguous: is this area or number?

**Answer: done**

Line 257-258, This depends on the status of the glaciers, see e.g. Pellicciotti et al., 2010. You can have a decrease in area and decrease in glacier melt.

**Answer:** the suggestion has been considered

Lines 258-259, avoid using two times 'However: : :'.

**Answer: done**

Line 261, ': : :extremely broad: : :' not clear to me what you mean here, use clearer/better word(s).

**Answer: done**

Line 284, replace 'These authors: : :' with 'They: : :'.

**Answer: done**

Lines 284-287, wrong word order, write 'They observed: : :'. Too long sentence, make two out of it.

**Answer: done**

Line 291, delete 'both'.

**Answer: done**

Line 296, write ': : :than the mean: : :'.

**Answer: done**

Line 298, write ': : : more than the: : :'.

**Answer: done**

Line 303, what do you mean with ': : :relevant: : :'? Try to be more clearly. Also: mentioning 'maximum monsoon temperature' and 'glacier melt' as main drivers of change is somehow redundant in my opinion, as the last is clearly directly dependent of the former one in your calculations. Maybe explain here better the dependencies.

**Answer:** we agree that it is redundant.

**Correction: Therefore temperature has been deleted from the PCA and the text modified accordingly.**

Lines 303-305, too long and complicated sentence, untangle and make two out of it.

**Answer: not done**. We did not understand the comment.

Line 315, write ': : :basin: : :'.

**Answer: done**

Line 317, maybe you can mention, that based on your findings it can be clearly seen, that glaciers act as buffers of the hydrological cycle.

**Answer:** glaciers are not the hydrological buffer, the glacier cover is the discriminant variable

**Correction: The concept has been added in the new version.**

Line 328, remove 'very' or use 'relatively'.

**Answer: done**

Line 330, write 'compare'.

**Answer: done**

Lines 333-335, wrong word order and too long sentence. Write 'The surface area of ponds-without glaciers strongly decreased (-25_6%, $p<0.001$) from 1963 to 2013. In contrast, the surface area of ponds-with-glaciers decreased much less (-6_2%, $p<0.05$) for the same period.'

Also: refer to Table 3 in that sentence.

**Answer: done**

Lines 361-362, contradiction to line 355 and Figure 9b., should be the other way round I suppose.

**Answer:** the comparison should be done with ponds without glaciers (line 354).

**Correction: we inserted the reference figures and type of lakes.**

Lines 362-363, here you could think about glacier morphology to further explain differences in glacier melt at different elevations (area, steepness, debris), if this is valid in your case study.

**Answer:** see the comment above

Line 369, be more precise when using the term 'glacial ponds' in order to separate them from supraglacial ponds etc.

**Answer: done**

Line 372, missing word(s) in 'The continued shrinkage of glaciers likely due to: : :'.

**Answer: done**

Line 376, avoid using 'study' two times.

**Answer: done**

Line 377, I wonder if the behavior of precipitation and glacier melt can be detected separately based on tracked pond areas. Maybe you can state something about this here.

**Answer: done**

Lines 382-387 & lines 389-391, did you directly observe constant (until the 1990s) or reduced glacier melt (in the early 2000s) or is this assumption based on the decreased max. air temperatures? It would be good if you could add here more background from your findings.

**Answer:** the concept has been clarified.

**Correction: through the analysis of surface area changes of unconnected glacial ponds.**

Line 403, write ': : :other climatic: : :'.

**Answer: done**

Line 409, verb missing.

**Answer: done**

Technical corrections (tables/figures):

Table 2:

Line 629, write ': : :of all considered: : :'.

Pond area, rounding error for max. value in 2nd and 3rd column (56.3 vs. 56.2)?

Basin, maybe you can add once in the paper how the basin is defined (='hydrological' catchment?) and how you calculated it (algorithm?).

Basin aspect, did you consider the calculation for directional values? Mean, median, range etc. of aspects have to be derived carefully, as e.g. the mean and median of the three values 45_, 345_ and 360_ doesn't make sense if calculated normally. Add a short note how you deal with this once in the paper where 'aspect' occurs first.

Also: How did you derive the mean basin aspect? Add used method ('vectorial mean').

Glacier aspect, same as 'basin aspect', see comment above. Here it seems that the median is not within the range.

**Answer:** "Hydrological basin" has been inserted in many key points of the manuscript.

The errors have been corrected. The method used for deriving the mean, median, etc.. of aspect has been described. The hydrological basin has been delineated with ArcGIS® hydrology tools.

**Correction: the circular statistic has been used for computing the (vector) mean and median values of glaciers and basins aspect (Fisher, 1993). The delineation method has been described.**

Table 3:

Asterisks, what do they stand for? Statistical significance level? Add explanation.

**Answer: done**

Table 4:

Basin aspect, again, how did you calculate mean and median basin aspect(s)? Asterisks, what do they stand for? Add explanation.

**Answer: (see the answer above), done**

Figure 1:

Line 684, you could add the source of the two pictures.

**Answer: done**

Figure 2:

a), use decimal degrees as written in text (line 91).

Also: black triangle and 'SNP' somehow misleading in inset map.

b), write ': : :isotherms corresponding: : :'.

Also: write 'max. temperature'

Line 715, remove ': : :'.

**Answer: done (point a: we changed the text)**

Figure 4:

Low image quality, especially axis labels. Try to improve.

Also: change x-axis labels to more 'intuitive' years, e.g. 1980, 1985,: : : and add year labels to all subplots a-d for better readability.

b), write 'Precipitation (anomaly)'

**Answer: done**

Figure 6:

Low quality, labels and lines.

Also: units missing.

a), y-range seems to be too small, missing points.

Also: wrong labels both at y-axis and in legend ('cumulate').

b), the left and right y-axes seem to be shifted vertically.

Line 777, a) and b) mixed?

Line 779, write ': : :Figures: : :'.

**Answer: done**, units in the caption

Figure 7:

Especially subplots a) and c) too small.

Also: size of circles in subplots b) and d) not clear, explanation below not clear as well.

Line 783, write 'Increased pond surface areas' and 'Decreased pond surface areas'.

Lines 785-786, description of subplots a) and c) not consistent with actual titles in plot (with/without glaciers).

**Answer: done**

Figure 8:

Add units for right y-axes (precipitation, melt). Also: make lines and bars in both sub plots identifyable, label them.

**Answer: done**, units in the caption

Figure 9:

Low quality, too small (axes labels).

**Answer: done**

Technical corrections (supporting information):

Figure SI1:

Last sentence in caption: write 'In Table 1 the relevant coefficients of correlation are reported.'.

**Answer: done**

Figure SI2:

a), add more space in between x-axis-labels. b), change x-axis-labels to more 'intuitive' years (e.g. 1980, 1985, : : :).

**Answer: done**

Figure SI3:

Very low quality of all labels, axes, wrong number of digits etc., too small. Also: add units or write that the anomalies are relative or dimensionless.

**Answer: done**

Figure SI4:

Low quality of all labels, too small. Second last sentence in caption: write ': : :considering Tmax and Tmean.'.

**Answer: done**

References:

Biggs, J., P. Williams, M. Whitfield, P. Nicolet and A.Weatherby, 2005. 15 years of pond assessment in Britain: results and lessons learned from the work of Pond Conservation. Aquatic Conservation: Marine and Freshwater Ecosystems 15: 693–714.

Pellicciotti, F., A. Bauder and M. Parola. Effect of glaciers on streamflow trends in the Swiss Alps. Water Resources Research, 46: W10522.

Watson, C.S., D.J. Quincey, J.L. Carrivick and M.W. Smith, 2016. The dynamics of supraglacial water storage in the Everest region, central Himalaya. Global and Planetary Change 142: 14–27.

---

## Author Response (AR1)

**General comments**

Understanding the links between climate, glaciers and hydrology in high mountain area is a growing and very important topic. This paper builds on other work by this group. There is potentially an interesting paper in here, which is novel and might lead the way to demonstrating how the changing size of ponds in mountainous regions that are not in immediate contact with ice but which contain glaciers in their catchments might be used to infer spatial and temporal trends in climate (precipitation, temperature, evaporation, glacier melt). The paper uses a statistical approach to the problem and the authors are to be commended for such a detailed analysis. Eventually one might imagine being able to use perhaps a more robust physically based approach, similar to that used by, e.g., Leclercq & Oerlemans, to reconstruct climate from glacier length fluctuations. This paper could be a useful stepping stone in that direction. [P.W. Leclercq, J. Oerlemans 2012. Global and hemispheric temperature reconstruction from glacier length fluctuations 38 1065-1079, doi: 10.1007/s00382-011-1145-7]

**Comment:** we thanks the reviewer for the detailed revision of the paper. Generally, we hope that the readability is now improved and the key messages are emerged.

I see 4 key problems with the paper as it currently stands although I hope the authors might be able to deal with these, re-orientate, focus, correct things and rewrite the paper so that it provides a better contribution to the cryospheric sciences.

1. The aim, objectives and overall general methodology of the paper are not articulated towards the beginning of the paper, so that the reader [or this one at least] remains generally confused about what is being done and, more importantly, why things are being done and has to gradually piece things together while reading the paper.

**Comment:** more specific objectives have been inserted. The overall general methodology has been described in a specific new paragraph.

2. The paper is very involved and dense with lots of different levels of analyses, and lacks a clear focus of what it is trying to achieve. I'd encourage the authors to work out what the key take home messages of the paper are and to present only the material that leads to those conclusions.

**Comment:** we hope that after having described the overall methodology the reasons behind the analyses could be emerged.

3. The paper is hard to follow, with sufficient ambiguities, inconsistencies, apparent contradictions and small lapses in grammar and syntax, to justify rewriting quite large sections, especially the Abstract and Conclusions. It would benefit from running through a spell checker and from proof reading by a native English speaker if at all possible.

**Comment:** the abstract and conclusion have been largely rewritten. Considering we are not native English speaker, before submitting the last version of the paper, we provided to submit the paper to the

American Journal Expert for the proof reading. An expensive certificate was released. We hope the kind help of the three reviewers could have deleted the grammar and syntax errors.

4. I query some of the scientific assumptions / results **Comment:** please read the answers reported below.

I elaborate on these points below.

**Specific comments**

1. The paper needs to articulate what the overall aims, objectives and methodology are. Currently, all we have on lines 83-86 is this: "This contribution examines the surface area changes of unconnected glacial ponds on the south side of Mt. Everest (an example is shown in Figure 1) during the last fifty years to evaluate whether they act as potential indicators of changes in the main components of the hydrological cycle (precipitation, glacier melting, and evapotranspiration) at high elevations in the Himalayan range." Even as a general aim, this is rather vague. This needs tightening up, we need to be given some more specific objectives and told an overall methodology of how these objectives will be achieved. Currently, after these 5 lines, we have an introduction to the field area (Section 2) followed by a detailed section on Data and Methods (Section 3). But when reading Section 3, we don't know why we're being told about the climate data, digitization of ponds, calculation of glacier surface area and melt, derivation of morphological parameters , etc. For example, on line 203 you refer to "degree of correlation among the data" But we have no idea what precise data you're talking about, nor why you want to correlate them.

**Comment:** More specific objectives have been inserted. The overall general methodology has been described.

- 2. The paper is very detailed, convoluted and involved, with a lot of separate components:
- i) looking at correlations between reanalysis climate data and ground climate data after 1994 to see which reanalysis products may most reliably be used to infer climate in the region prior to 1994;
- ii) generating other proxy data ultimately from the climate data, notably evapotranspiration and glacier melt (using a simple temperature index model);
- iii) calculating glacier shrinkage and "unconnected pond" area shrinkage (where "unconnected ponds" refer to those not physically in contact with glacier ice) for 6 time periods since 1963 from a map (1963) and satellite imagery (1992, 2000, 2008, 2011,2013);
- iv) performing a suite of non-parametric statistical tests to investigate whether trends in pond area, glacier area, climate & climate derivatives (evapotranspiration and glacier melt) are statistically significant in different time periods (e.g. the whole period 1963-2013 or sub-periods 1963-1992, 1992-2013); between different types of unconnected pond (those whose upstream catchment is > 10% or < 10% glacierised) or for different "morphological boundary conditions" (e.g. elevation, aspect);</li>
- v) performing a Principal Components Analysis on the variables to investigate climate drivers of pond area change.

Furthermore, some of the analysis is done on the full set of 64 ponds, and some is done on a sub-set of 10 ponds. Similarly, some of the analysis splits the time period into two (1963-1992 and 1992-2013) and some

splits the time period into three (1963-1992, 1992-2000 and 2000-2013). All in all, the reader gets rather bogged down in the detailed analysis and loses a sense of the big picture.

**Comment:** we thanks the reviewer for this tentative of summary. We used this scheme for generating a paragraph related to the overall methodology.

3. Because the paper has many different strands, it is particularly important to have a very clear abstract and conclusion. Reading the abstract, it is not at all clear what the key take home messages of the work are. Unfortunately, having ploughed my way through the paper and emerged somewhat exhausted from the final sentence of the conclusions, I was still rather unsure what the key conclusions were.

Lines 369-371 tell us that during the monsoon period the "unconnected ponds" declined in area (by 10%). Fine, this is clear.

Lines 371-372 tell us that this is due to a drop in precipitation and a decrease in maximum temperature (and therefore glacier melt). Also quite clear.

Then it gets confusing. Lines 372-373 tell us that "the continued shrinkage of glaciers likely due to the effects of less precipitation than an increase in temperature". This is not a grammatically correct sentence but I assume the authors mean that "the continued shrinkage of glaciers [is] likely due to the effects of less precipitation [rather] than an increase in temperature." I don't recall where in the paper this was discussed. The paper involved a statistical analysis explaining variation in pond area not glacier area. By "continued shrinkage" I assume the authors are referring to the actual shrinkage that occurred in the past, and are not speculating about shrinkage that may or may not occur in the near future? Note how we're told that pond area shrinkage is due to a "decrease in maximum temperatures" but that glacier shrinkage is likely not due to an "increase in temperature". It's a little ambiguous whether temperature decreased, although not significantly. On line 281 we're told that maximum temperatures decreased. On line 282 we're told that minimum temperatures increased. Actually we're told that the increase in the minimum temperature "balanced" the decrease in the maximum temperature, although this isn't strictly correct as then, I assume, the mean would stay exactly the same. Is it really the case that mean temperature decreased? Figure 4a, shows that the mean temperature increased over the time period!

**Comment:** we provided to underline the key messages; the main conclusions have been rewritten and clarified; Figure 4a shows the trend of the mean annual temperature (which is increasing). Lines from 280 to 282, as specified in the text, report the trends during the monsoon period (the mean temp is slightly decreasing). However, in general, we accept the general suggestion that the discussion is too much convoluted. Therefore our efforts were devoted to simply the discussion.

Section 4.3 is virtually impossible to follow. It spans just a side of A4 during which we're asked to study Table 4, then Table 2, then Fig SI3, Table 3 and Figure 4. That's just the first short paragraph. We then need to look at Fig 5, SI4 and SI5, Fig 6a and SI4, back to 6b, back to SI4, then again, and again, then flip back to 2b. We then have to jump forward again to 6b, move to Table SI5, Figure 5, and Fig 5 again, Table 4, Figure 6 and finally back to Table 4.

I was concerned throughout this section that I was moving the pages back and forth so much that I'd accidentally end up making some sort of 3D origami animal. I'd encourage the authors to cut down on the Figures and Tables and discuss things in a way that doesn't involve so much movement.

**Comment:** we tried to simply this section.

4-1. Can you explain better how melt is being derived for the glaciers? In lines 171-176, is it necessary to refer to the work of Salerno et al (2015) regarding the calculation of temperature at the mean elevation of each glacier? Is it not the case that the pyramid data are used together with a lapse rate (tell us what the lapse rate is) and the melt factor to calculate the melt across each elevation band (tell us what the band width is and what DEM is used) and that these are then summed for each glacier to calculate the melt to each glacier?

Correction: the text has been corrected according to the suggestion.

4-2. Given the way that you're calculating glacier melt, there will be huge autocorrelation between Tmax and Glacier melt. So it's not surprising that your correlation coefficients involving Tmax and Glacier melt are so similar. I'm therefore surprised by Fig 5 where you seem to show that glacier melt and Tmax are two strong independent variables contributing to the principle components. Have I understood this correctly?

**Comment:** The PCA shown in Figure 5 attempts to provide an overall overview of the relationships among the trends related to the potential drivers of change and the pond surface areas: glacier melt and precipitation, while evaporation is excluded. Following the suggestion Tmax probably needs to be removed to avoid that the reader could think that our aim is to show similarities between Tmax and melt (Tmax).

Correction: Tmax has been removed from the PCA.

4-3. Table SI5. Do I understand this analysis correctly? For each pond, are you only working with 14 data points? Is this sufficient to demonstrate every variable is normally distributed so that you can use the parametric correlation test (as you state you do lines 203-5)

**Answer:** Yes the interpretation is correct. We used the annual ponds surface area for the 2000-2013 period and we compared the area with the correspondent driver of change (14 comparisons). The number of years considered in the analysis is given by the availability of satellite imagery. Given a not so much elevated number of comparisons, however, we need considered that the same analysis is repeated for (corroborated by) 10 lakes which present very similar relationships with the selected variables. No other data is available for the past.

Moreover, to test the normality of the comparisons there is not a minimum number of data. Razali and Waph, 2011 demonstrate that the Shapiro-Wilk test (used in this paper) presents the highest power for small sample size (analyzing sample size ranging from 10 to 2000).

**Correction:** we wrote in the text that Razali and Waph, 2011 demonstrate that the Shapiro-Wilk test presents the highest power for small sample size.

4-4.k On line 100 you tell us that the precipitation has a specific gradient. Given that you go to all the trouble of calculating glacier melt using a lapse rate, and given the importance of precipitation for your analysis, why do you not use this lapse rate in the calculation of precipitation from the pyramid station when analysing the precipitation relevant to the different ponds? The ponds are at different elevations, and the catchments above them have different elevation ranges (and hypsometries). The pptn gradient above 2500m is non-linear. All these things will mean the precipitation falling above the lakes in your analysis will be very different for the different lakes.

**Comment:** In this analysis we are not interested in the absolute (annual cumulate) value of precipitation on each specific ponds. If it was this case, as suggested by the reviewer, applying the precipitation

gradient analyzed by Salerno et al., 2015, we could be able to estimate it. In order to analyze the possible relationships between pond surface area changes and precipitation variations we need to compare the just the trends of these variables. Therefore, 10 ponds were selected and their surface areas tracked yearly. For each pond, the series of annual surface areas has been compared vs annual precipitation series. We carried out the same procedure for the glacier melt. The assumption behind this analysis is that the precipitation trend along the gradient and along the valleys is the same. This is a reasonable assumption/limitation due to the fact that land precipitation series at this elevation are so rare. However, the last paragraph aims to investigate this assumption: the result is that there is not an altitudinal or spatial pattern.

Correction: the assumption has been specified in the text, as well as, its analysis in the last paragraph.

4-5. Section 3.5. I'd like to see a better articulation of the sources of error and how they were calculated for this study. First you imply error is a function of linear error and perimeter. Then you refer to a linear resolution error and a co-registration error. This all needs explaining more carefully and precisely.

Answer: we applied this procedures in other papers, probably here was too much hermetic.

**Correction: the paragraph has been rewritten.**

Technical corrections; typing errors, etc.

There are a lot and I don't have time to give them all. Below I give some of the key ones. Numbers refer to line numbers.

14. "unconnected ponds" This is defined in the paper but the abstract should be intelligible on its own. Explain what is meant here.

**Answer: done**

15. "We infer an: : :"

**Answer: done**

17-19. Rewrite. I think this should be at least 2 sentences. Meaning not at all clear.

- Answer: done
- 31. glacier

**Answer: done**

44. ": : : increases in the evaporation / precipitation ratio: : :" [refer to evaporation / precipitation ratio also above on line 41 to be consistent]

**Answer: done**

51-53. Vague. Rewrite.

**Answer: done**

61. What do you mean by "these lakes"? Just proglacial lakes or all 3 categories?

**Answer: done**

64. "decidedly similar". To what?

**Answer: done**

67 opening

**Answer: done**

67. Ref to englacial conduits is relevant to supraglacial lakes but not proglacial.

**Answer: done**

54-72. Para could be shorter with tighter articulation of key relevant points.

73 A valuable

**Answer: done**

75 glacierized not glaciated.

**Answer: done**

75-6. ": : : region has the largest number of lakes in: : :"

**Answer: done**

78. reduced dimensions. Do you mean "relatively small size"?

**Answer: done**

80 ": : : make them especially: : :"

**Answer: done**

78-82. This sentence is confusing. Is it their small size that's relevant or the low water volumes and high surface area to depth ratios. You start the sentence implying it's the first, and end saying it's the 2nd & 3rd attribute that's important. Rewrite.

79. Can you check the entire document? Here you define lakes and ponds according to size. But earlier and later you use the terms interchangeably and (according to this definition) sometimes incorrectly. You need consistency. Define at the very start of the paper. You could use "water bodies" if you want a generic term.

**Answer: done**

89. Do you need the abbreviation "CH"? Do you use this term again?

**Answer: done**

93-4. ": : : of the territory contains temperate glaciers and less than 10% is forested."

**Answer: done**

97. "For the last 20 years" Avoid phrases like this. Later you refer to "the last decade" I think too. These phrases are ambiguous. The last 20 years means 1996-2016 to me, but actually pyramid station has been operating since 1994. Always state the precise dates to avoid confusion.

**Answer: done**

99 ": : : precipitation falls between June and Sept: : :"

**Answer: done**

102. ": : : :large glaciers in the SNP are: : :"

**Answer: done**

103. Delete "In the SNP"

**Answer: done**

109. "realised the complete cadaster" What does this mean?

**Answer: done**

110. "univocal" suggest change to "unique"

**Answer: done**

113 "::: Everest after the:::"

**Answer: done**

118. check grammer here.

**Answer: done**

122 ": : : and the monthly cumulated: : : "

123 delete "recently"

**Answer: done**

125. Why evapotranspiration not also calculated for 1994-2002?

**Answer: done**

126. "recorded continuously" Is this a monthly time-series too? Or calculated more frequently and averaged?

**Answer: done**

130. You casually say "before the 1990s" but you should say before 1994. See other instances of this throughout the paper,

**Answer: done**

143. "intermediate periods" is confusing. Why not just say "scenes"?

**Answer: done**

146. "environments" is completely the wrong word. Do you mean "biases?

**Answer: done**

147-8 "For the 2000 - 2013 period, due to the wider availability of satellite imagery, ten ponds were: : :"

**Answer: done**

155. Semester is the wrong word

**Answer: done**

158. "these characteristics" What characteristics are you talking about here?

**Answer: done**

161. "The acceleration disappears" This is wrong. No acceleration has been discussed

**Answer: done**

previously. Do you mean that there is a decrease in area?

**Answer: done**

167. "pond basins" This is a bit unclear. You're referring to the basins (or catchments) containing? Or Upsteam of? The ponds.

**Answer: done**

178. remove the phrase "such". Just list all the parameters you use.

**Answer: done**

180-181. Vertical accuracy greater than horizontal? Are you sure?

Answer: yes we have checked, please refer to Tachikawa et al., 2011.

185. Is this EM also used for defining the elevation bands for the calculation of melt? Should have been referred to earlier.

**Answer: done**

187. Map not maps.

**Answer: done**

194. morphological? Or best to use morphometric for consistency.

**Answer: done**

217 pond size

**Answer: done**

221 before 1994

223. Why are seasonal data shown for temperature but not precipitation in Table 1?

**Answer:** during the monsoon, as described in the text, the precipitation are the 90% of the annual cumulated amount. Therefore outside the summer, during the pre and post monsoon season, the seasonal cumulated amounts are often equal to zero. Thus the parametric statistic does not make sense. We decided to present the data aggregated at annual level, as compromise.

235. Are the 170 ponds all from the SNP region?

**Answer: done**

237. delete "prefer to"

**Answer: done**

238. "environments"? Do you mean ponds? Water bodies?

**Answer: done**

235-242. You don't refer to columns 1 & 2 in Table 2. Are these redundant? Remove them?

**Answer:** we think that the two columns are important and cannot be removed because they point out the different features of the two groups of data.

248. "glacier surface differences" ? Do you mean glacier surface area changes?

**Answer: done**

250. Further loss of area (-18%) is ambiguous. It's not an extra 18% loss since 2011.

**Answer: done**

251. Poor grammer

**Answer: done**

255 "Having analysed: : :"

**Answer: done**

257. delete "Usually and"

**Answer: done**

258. "this inbound component" Do you mean glacier melt input?

**Answer: done**

259-264. Vague, confusing and poor English here.

**Answer: done**

302. don't need the word "monsoon" at the end of this line with reference to temperature

here do you? All these variables are for the monsoon right?

**Answer: done**

303 "relevant" is the wrong word

**Answer: done**

307 "sensible factor" is incorrect.

**Answer: done**

322 ": : : : ponds were in catchments with a glacier: : : "

**Answer: done**

323-3. Needs writing.

**Answer: done**

324-5. Why are you calling ponds in catchments that are

Correction: this methodological aspect has been inserted.

3) As acknowledged by the authors the study area is full of debris covered glaciers. The applicability of the glacier melt model used for debris covered glacier must be discussed.

**Answer:** The glaciers within the pond basins are not debris covered. In this region debris covered glaciers are usually glaciers of a certain size with a developed flat ablation area. In all considered pond basins, the glacier are very small, steep (31°), clings to the mountain peaks, without having developed debris covered ablation area.

Correction: following the suggestion of the reviewer we specified in the text these features of glaciers within the pond basins.

4) It is known that SOI toposheets derived from winter time areal imagaries may contain significant errors. Some of the authors have published results using high resolution Corona KH4 images from 1962 in this area. Could the same images be used to verify the baseline 1962 extents of the ponds studied? Corona data should help in filling the large time gap between 1962 and 1992.

**Answer:** We did not used only Corona image for digitalizing all 64 considered ponds because many of them in this image are snow-covered, but, we checked the quality of the map comparing the size of some ponds in both data sources.

Correction: "The topographic map of the Indian survey of 1963 (hereafter TISmap-63, scale 1:50,000) was used to complement the results achieved using the declassified Corona KH-4 (15 Dec 1962, spatial resolution 8 m). Thakuri et al., 2014 describe the co-registration and rectification procedures applied to the Corona KH-4 imagery. Unfortunately, on these satellite images many ponds are snow-covered. Therefore here the ponds surface area digitalized on TISmap-63. The accuracy of this map has been tested comparing the surface areas of 13 ponds digitalized on both data sources (favoring the cloud and shadow free ponds). Figure SI1 shows the proper correspondence of these comparisons. Furthermore, in order to estimate the mean bias associated with TISmap-63, we calculated the mean absolute error (MAE) (Willmott and Matsuura, 2005) between data, which resulted sufficiently low (3.6%), assuring in this way the accuracy of ponds surface area digitalized on TISmap-63."

5) Which climate data is used for the correlation studies? Pyramid data or reanalysis/ gridded products? If pyramid data is used then what is need of describing the others? If the gridded/reanalysis data are used then why not study the correlations for a period longer than the time-window of 2000-2013? What happens if the analysis is extended to all the ponds and for the duration of 1962-2013 using the GPCC precipitation data?

**Answer:** As described above the correlation studies have been done using the Pyramid data due to the continuous series of annual ponds surface area are available only for the 2000-2013 period and land meteorological data are available for 1994-2013 period. This explains, answering to the reviewer, why a time-window longer than 2000-2013 does not exist. Extending the analysis to the 1962-2013 is not possible because before 2000 we have just two years in which it was possible to digitalize the ponds (1963 and 1992).

Gridded/reanalysis data are not used here for correlation studies, but to obtain information, as written in the paper, on climatic trends in the antecedent period (before 1994). For this reason they have been compared with land data and the best products have been chosen.

Correction: following this suggestion and all other comments received in particular from reviewer 1 a need of clarification clearly emerges. Therefore the method section of the paper has been restructured and these concepts clarified.

6) The details of the computation of the mean pond area change and its uncertainty may be explicitly pointed out.

**Answer:** as even suggested by another reviewer the section related the uncertainty computation is too hermetic.

**Correction: Therefore it has been rewritten.**

7) While the authors do a good job of pointing the reader to the appropriate references, at times they may become distractions. For example while both the following cited references are great read in their own ritght, the citations here may be a bit far-fetched - "The current study is focused on the southern Koshi (KO) Basin, which is located in the eastern part of central Himalaya (CH) (Yao et al., 2012; Thakuri et al., 2014) (Fig. 2)". Also refer to Major comment (2) in this context.

**Correction: we deleted Yao et al., 2012**

8) How are the periods of 1992-2000, 2000-2008, 2008-2011 and 2011-2013 used in table 3 selected? **Answer:** the periods have been selected in relation to the availability of satellite imagery

9) The conclusion has lengthy discussions about glacier changes and only a few words on the multi-temporal pond extent data described in the rest of the paper. The connection between the claimed signal from pond area change and glacier changes in the region is not explicitly mentioned as well.

Answer: the conclusions have been rewritten and the "connection" has been more explicitly mentioned

10) Some typographical errors: 1 67 "opeping" 1 122 : "montly comulated" 1 194: morphologicalal **Answer:** The suggestion has been followed

**P. Buri (Referee)**

**Summary:**

The authors investigate surface area changes of ponds over a period of fifty years (1963-2013) in a highelevation Himalayan region using a topographic map (1963) as well as various Landsat satellite images (1992-2013). They relate the observed area changes to precipitation, temperature and glacier melt trends. The meteorological dataset used in this study is based both on a high-elevation weather station in the catchment (operating since the mid 1990's) and regional gridded and reanalysis data used to extend the record back in time to the 1960's, for which the authors have the first inventory of ponds (1963). The authors find a high sensitivity of ponds to a change in climate and try to use water bodies as proxies to detect behavior of precipitation and glacier melt.

**General comments:**

The paper is generally well written and structured in a clear way. However, I have some major issues regarding the methods applied that question partly your conclusions. In addressing these points (mentioned below) the paper may could be improved considerably and your original dataset and conclusions could be presented in a concise way and more scientific value could be added to your work. You relate changes in the climate to changes in the lake areas, as meteorological parameters are often represented in a highly limited way in remote and high-elevation regions. This is an interesting but also novel concept and addresses a relevant scientific question within the scope of the journal, as e.g. temperature and precipitation build the base for many research questions in various fields of the cryosphere. However, it is questionable if the approach used in this study can be used to reconstruct changes in the climate as lakes respond to many inputs as say yourself, so pond area is only an integrated variable (see point 4 below). The provided references appropriate and referenced in a helpful way in the text. At least one new study (published after submission of this manuscript, see major point 1 below) should be added. The statistical analysis and the results, respectively, are not fully clear everywhere in the manuscript (e.g. Table 3, see point 3 below). The methods description is rather complete, with methods explained either directly in the text or by referring the reader to further literature. They major issues to address are listed here:

**Comment:** we thanks the reviewer for the revision of the paper. Generally, we hope that in the new version the key messages could emerge more clearly. All the suggestions have been followed. A new overall methodological section have been introduced.

Major issues:

1) Satellite images used for the analysis:

First, you need to indicate in the main text, including abstract, which satellite images you use (not only in the supplement) as this is a key information. You use Landsat (from Table 2 of supplement) and there might be an issue of too coarse resolution with Landsat. Pond area strongly depends on the accuracy of the derived outlines. This is a key issue and you should provide some errors in your delineation, mainly due to the resolution of the images. Watson et al. (2016), looking at supra-glacial ponds though, show that resolution is an issue and they state that Landsat products cannot be used for this purpose. So may cite this paper (which

came out after your submission) and also consider that issue. Maybe your ponds are very big and not affected by the coarse resolution of Landsat? A clear advantage of Landsat is that it allows going back in time – what the higher resolution products cannot as they are all for recent years. Also, from Table 3 of supplement there is an ALOS image listed, although it is not clear what is that used to. ALOS has a different resolution and so this should be discussed.

**Answer:** As suggested by the reviewer the supraglacial lakes in Mt Everest Region are very small. According to Watson et al. (2016) their size range from 0.09 to 0.36  $10^4$  m2, while the unconnected ponds in the same region (this study) are on average 1.1  $10^4$  m2, i.e., an order of magnitude larger. This is not the unique difference between the two kind of ponds. As described in the text, supraglacial ponds are strictly connected with glacier dynamics, thus, as describe by many authors (and by the same Watson et al. (2016)) their measurement is very uncertain. Landsat imagery is surely too coarse for these ponds.

Considering unconnected ponds, in general, we tracked the pond surface changes in many papers (Tartari et al., 2008, Thakuri et al., 2015; Salerno et al., 2012, Salerno et al., 2014). We wrote a specific work (Salerno et al., 2012), on the uncertainty related to the measurements of lakes from satellite imagery in the region, which is referenced also by Watson et al. (2016). In the methodological section there is a section devoted to the uncertainty of measurements.

Table 3 is a general summary of surface area changes related to all 64 considered ponds glaciers located within the basins. In the previous version of the paper was not explicitly written that the same table reported the uncertainty of measurements. This could have confused the reviewer, which though that we did not consider and discuss the uncertainty of measurements.

The ALOS was used to track the pond surface areas in 2008, this image was preferred considering the better resolution. In fact in table these period presents uncertainties slightly lower.

Correction: 1) the methodological section related to the uncertainty of measurements has been extended. 2) we corrected the caption of Table 3. Along the paper, where it was omitted, the uncertainty has been associated with relevant difference of measurement. The satellite images used for the analysis have been also reported in the main text and in the abstract.

2) Degree-day model for glacier melt:

The use of a degree-day model for glacier melt might be a key limitation, as this has been shown to be very sensitive to temperature fluctuations. Therefore the estimates of "glacier melt" might be erroneous, and responding too much to changes in temperature. I would suggest that you perform calculations with a better model. Also, a key concern is that you use a constant melt factor from another study - the model needs calibration. If you cannot do this, you should perform an uncertainty analysis by varying this factor in a given range. In addition, why did you only use one factor and not two for snow and ice? I would strongly recommend that you: 1. do an uncertainty analysis and see how sensitive your results are to changes in the degree-day factor 2. use a more appropriate model

**Answer:** This paper does not aim to provide an accurate estimation of the magnitude of the melt released from glaciers located in the pond basins. In fact, its value has never been discussed and mentioned. The melt factor could be unsuitable, but if it was wrong no analysis would be compromised. We compared its 2000-2013 trend vs the pond surface areas, and the correlation analysis is independent from the magnitude of the compared series. Consequently, we do not need different factors for snow and ice and to make a sensitive analysis.

Being interested in the melt trend and not in its absolute magnitude and considering that these small glaciers are ungauged, we do not more sophisticated melt models, which consider specific geometries and differentiated melt factors. We are aware of the autocorrelation between the maximum temperature and glaciers melt calculated from this variables, i.e., their fluctuation are similar. The added value is only due to that the positive temperature calculated for each glacier (elevation bends) are able to generate a melt, which we found to be significant related to the observed pond surface area changes. If ponds (and glaciers) were located some hundred of meters at higher elevation, surely the melt and Tmax would be less correlated and the application of the degree-day model would look less trivial. What is the knowledge contribution of the application of the degree-day model in this contest? Maximum temperature trend is here demonstrated to be responsible of processes able to modify the pond surface area. How processes? Glacier melt is a reasonable factor, due to we find significant relationships when glaciers are present in the pond basins, and no relationship with Tmax when glacier are not present in the basin.

**Correction: these concepts has been inserted in the text.**

**3) Table 3:**

There are some very contrasting changes and it is not entirely clear how these values were derived: e.g. for ponds with glacier coverage

**Correction: the circular statistic has been used for computing the (vector) mean and median values**

of glaciers and basins aspect (Fisher, 1993). The delineation method has been described.

Table 3:

Asterisks, what do they stand for? Statistical significance level? Add explanation.

**Answer: done**

Table 4:

Basin aspect, again, how did you calculate mean and median basin aspect(s)? Asterisks, what do they stand for? Add explanation.

Answer: (see the answer above), done

Figure 1:

Line 684, you could add the source of the two pictures.

**Answer: done**

Figure 2:

a), use decimal degrees as written in text (line 91).

Also: black triangle and 'SNP' somehow misleading in inset map.

b), write ':: : isotherms corresponding: : :'.

Also: write 'max. temperature'

Line 715, remove ': : : '.

**Answer: done (point a: we changed the text)**

Figure 4:

Low image quality, especially axis labels. Try to improve.

Also: change x-axis labels to more 'intuitive' years, e.g. 1980, 1985,: : : and add year

labels to all subplots a-d for better readability.

b), write 'Precipitation (anomaly)'

**Answer: done**

Figure 6:

Low quality, labels and lines.

Also: units missing.

a), y-range seems to be too small, missing points.

Also: wrong labels both at y-axis and in legend ('cumulate').

b), the left and right y-axes seem to be shifted vertically.

Line 777, a) and b) mixed?

Line 779, write ':::Figures:::'.

Answer: done, units in the caption

Figure 7:

Especially subplots a) and c) too small.

Also: size of circles in subplots b) and d) not clear, explanation below not clear as well.

Line 783, write 'Increased pond surface areas' and 'Decreased pond surface areas'.

Lines 785-786, description of subplots a) and c) not consistent with actual titles in plot

(with/without glaciers).

**Answer: done**

Figure 8:

Add units for right y-axes (precipitation, melt). Also: make lines and bars in both sub plots identifyable, label them.

Answer: done, units in the caption

Figure 9:

Low quality, too small (axes labels).

Answer: done

Technical corrections (supporting information):

Figure SI1:

Last sentence in caption: write 'In Table 1 the relevant coefficients of correlation are reported.'.

**Answer: done**

Figure SI2:

a), add more space in between x-axis-labels. b), change x-axis-labels to more 'intuitive' years (e.g. 1980, 1985, : : :).

**Answer: done**

Figure SI3:

Very low quality of all labels, axes, wrong number of digits etc., too small. Also: add units or write that the anomalies are relative or dimensionless.

**Answer: done**

Figure SI4:

Low quality of all labels, too small. Second last sentence in caption: write

:: : : considering Tmax and Tmean.'.

**Answer: done**

152 An inter-annual analysis has been carried out during the 2000-2013 period (hereafter we refer to this 153 analysis as "short-term inter-annual analysis"), considering the wide availability of satellite imagery in 154 this period, on some selected unconnected ponds (hereafter we refer to these ponds as "selected ponds") 155 to continuously track the inter-annual variations in surface area. This analysis aims to investigate the 156 possible drivers of change (precipitation, evaporation and glacier melt) considering the availability of 157 continuous series of annual pond surface areas on the one side, and climatic data from a land station 158 located in the area-on the other. The study has been carried out through a correlation analysis and a 159 Principal Component Analysis (PCA).

160 An inter-annual analysis has been carried out during-from 1963 to 2013 (hereafter we refer to this analysis as "long-term inter-annual analysis") on a wider unconnected pond population (hereafter we 161 162 refer to this population as "all considered ponds") and on glaciers located within their hydrological basin. 163 Two kinds of analyses have been carried out on this set of data: 1) Pond surface area changes have been related to certain morphological boundary conditions. This analysis allows to investigate on the factors 164 165 controlling the pond surface area changes. The significance of the observed differences has been 166 evaluated with specific statistical tests; 2) Pond surface area changes have been related to climatic data. 167 This analysis aims to point out the capability of unconnected ponds to infer on the detected drivers of 168 change also in the past when land climatic data did not exist. This study needed a preliminary analysis to

169 reconstruct the climatic trends before the year 1994. Selected regional gridded and reanalysis datasets
 170 have been compared with land weather data available for the 1994-2013 period.

**171 3.1-2 Climatic data-**

172 The monthly mean of daily maximum, minimum, and mean temperature and monthlymonthly 173 comulated cumulated precipitation time series used in this study have been recently reconstructed for the 174 elevation of the Pyramid Laboratory (5050 m a.s.l.) (Fig. 2) for the 1994-2013 period (Salerno et al., 175 2015). The potential evapotranspiration evaporation for the period (2003-2013) has been calculated by 176 applying the Jensen-and-Haise model (Jensen and Haise, 1963) using the mean daily air temperature and 177 daily solar radiation recorded continuously during this the 2003-2013 period at Pyramid Laboratory. The 178 Jensen-and-Haise model is considered to be one of the most suitable evaporation estimation methods for high elevations (e.g., Gardelle et al., 2011; Salerno et al., 2012). 179

180 To obtain information on climatic trends in the antecedent period (before the 1990s1994), we used 181 some regional gridded and reanalysis datasets. We selected the closest grid point to the location of the 182 Pyramid Laboratory, and all data were aggregated monthly to allow a comparison at the relevant time 183 scale. With respect to precipitation, we test the monthly correlation between the Pyramid data and the 184 GPCC (Global Precipitation Climatology Centre), APHRODITE (Asian Precipitation-Highly Resolved 185 Observational Data Integration Towards Evaluation of Water Resources), Era-Interim reanalysis of the 186 European Centre for Medium-Range Weather Forecasts (ECMWF), and CRU (Climate Research Unit -187 Time Series) datasets. For mean air temperature, we considered the Era-Interim, CRU, GHCN (Global 188 Historical Climatology Centre), and NCEP-CFS (National Centers for Environmental Prediction-Climate 189 Forecast System) datasets, whereas for maximum and minimum temperatures, we used the Era-Interim 190 and NCEP-CFS datasets (details on the gridded and reanalysis products are reported in Table SI1).

191 **3.2-3** Pond digitization.

**192 3.3.1 Long-term inter-annual analysis**

193 Pond surface areas were manually identified and digitized using a topographic map from 1963 and 194 more recent satellite imagery from 1992 to 2013. The topographic map of the Indian survey of the year 195 1963 (hereafter TISmap-63, scale 1:50,000) was used to complement the results achieved using obtained 196 from the declassified Corona KH-4 (15 Dec 1962, spatial resolution 8 m). Thakuri et al., (2014) described 197 the co-registration and rectification procedures applied to the Corona KH-4 imagery. Unfortunately, on 198 these satellite images many ponds are snow-covered. Therefore here we considered the ponds surface area 199 digitalized on TISmap-63. The accuracy of this map has been tested comparing the surface areas of 13 200 ponds digitalized on both data sources (favouring the cloud and shadow free ponds). Figure SI1 shows the 201 proper correspondence of these comparisons. Furthermore, in order to estimate the mean bias associated 202 with TISmap-63, we calculated the mean absolute error (MAE) (Willmott and Matsuura, 2005) between 203 data, which resulted sufficiently low (3.6%), assuring in this way the accuracy of ponds surface area 204 digitalized on TISmap-63. 205 In total, five intermediate periodsscenes (details on data sources are provided in Table SI2) were

considered according to the availability of satellite imagery.-Landsat images have been mainly used,
 except in 2008, when in the region the ALOS image, presenting a better resolution, was
 availabilityavailable (details on data sources are provided in Table SI2).

209 We selected tracked only those ponds present continuously in all these five periods to exclude possible

ephemeral environmentswater bodies. As described below, 64 ponds haves been tracked from 1963 to
 2013 (Fig. 2a).

**212 3.3.2 Short-term inter-annual analysis**

From the 2000-(2000-2013 period), due to a wider availability of satellite imagery (and in particular the Landsat imagery) in the region for the last decade, 10 ponds were selected among the pond population (64 ponds) considered in the long-term analysis (1963-2013) to continuously track the inter-annual variations in surface area in the recent years. The largest ponds, free from cloud cover, and with diverse glacier coverages (from 1% to 32%) within their hydrological basin were favored in the selection (details on data sources used for these lakes-ponds are provided in Table SI3).

219 3.3.3 Intra-annual analysis

220 The intra-annual variability in pond surface area has been investigated throughout the year 2001 221 through the availability of 5 cloud-free satellite images from June to December (details on data sources 222 used for these lakes ponds are provided in Table SI4). The first semester months of the year was were 223 excluded from the analysis because many ponds were frozen until April/May. Even in this case, the main 224 criterion drovedriving the ponds selection was the ponds were selected based on the absence of cloud 225 cover from the satellite images over the pixels representing the pond surface area. Only ponds for which a 226 continuous series of data was tracked retrieved from June to December were selected. Moreover for all 227 images, and the largest lakes ponds to the reduce the uncertainty in the shoreline delineation with various 228 degrees of glacial coverage-were favored in order to reduce the uncertainty in the shoreline delineation. Thus, 4 lakes ponds with these characteristics were selected, and their intra-annual variability is tracked in 229 230 Figure 3. Based on Figure 3, Wwe observe a common significant increase in pond surface area during the 231 summer months, likely due to monsoon precipitation and high glacier melting rates. The acceleration This 232 increase in surface area disappears on average during the fall. Some single ponds present a dispersion of around 5% between October and December (LCN4 and LCN77). However, the same Figure points out 233 234 that just averaging this information on a population only a little bit larger, the dispersion between October 235 and December becomes almost zero (1%). Therefore these months are period from October to December 236 is the best period to select the satellite images necessary for the inter-annual analysis of pond surface area. 237 In fact, during these months, the ponds are not yet frozen, the sky is almost free from cloud cover, and, as 238 observed in Figure 3, the inter-annual analysis on average is not affected by intra-annual seasonality. 239 Consequently all images for the inter-annual analysis have been selected from these months (Table SII; 240 Table SI2). Generally, climatic inferences coming from the analysis of surface area of ponds surely needs 241 to consider a wider number of ponds in order to reduce the intra-annual variability due to the local 242 conditions of each lake.

**243 3.3-4 Glacier surface areas and melt.**

Glacier surface areas within the pond-basins containing the ponds were derived from the Landsat 8
remote imagery (October 10, 2013) taken by the Operational Land Imager (OLI) with a resolution of 15
m. The satellite imagery used to trace-track the inter-annual variations in glaciers since the early 1960s is
reported in Table SI2. Detailed information of digitization methods are described in Thakuri et al., 2014.
To simulate the daily melting of the glaciers associated with the 10 selected ponds, we used a simple

249 T-index model (Hock, 2003). This model is able to generate daily melting discharges as a function of

daily air temperature above zero, the glacier elevation bands (using the Digital Elevation Model –DEMdescribed below)7, and a melt factor (0.0087 m d-1 °C-1) provided by Kayastha et al. (20082000) from a
field study (Glacier AX010) located close to the SNP (southwest). The Glacier AX010 glacier is a small
debris free glacier, located in the Dudh Koshi valley in same climatic and geographic setting of glaciers
considered here.

255The choice of using a simple model of melting is due to the fact that this paper does not have the256specific objective to provide an accurate evaluation of the magnitude of the melt water released from257glaciers located in the pond basins, but rather to estimate its trend, as function of the temperature, in order258to evaluate if the glacier melt is a possible driver of changes of the pond surface areas. Being interested in259the melt trend and not in its absolute magnitude and considering that these small glaciers are ungauged,260we do not need more sophisticated melt models, which consider specific geometries and differentiated261melt factors.

262 T-index model has been applied here considering the daily temperature of the Pyramid Laboratory
 263 corrected using the monthly lapse rates reported in Salerno et al., 2015 for each 50 m glacier elevation
 264 band. The melt estimated for each band has been then summed to calculate the total melt realized byfor
 265 each glacier.

**266 3.4-5 Morphometric parameters**

267 The parameters related to the ponds basin as the area, slope, aspect, and elevation were calculated 268 through the Digital Elevation Model (DEM) derived from the ASTER GDEM (Tachikawa et al., 269 2011). The ASTER GDEM tiles for the Mt. Everest region were downloaded from ...-The vertical and 270 horizontal accuracy of the GDEM are ~20 m and ~30 m, respectively (Tachikawa et al., 2011; Hengl and 271 Reuter, 2011). We decided to use the ASTER GDEM instead of the Shuttle Radar Topography Mission 272 (SRTM) DEM considering the higher resolution (30 m and 90 m, respectively) and the large data gaps of 273 the SRTM DEM in this study area (Bolch et al., 2011). Furthermore, the ASTER GDEM shows better 274 performance in mountainous terrains (Frey et al., 2012). Hydrological basins have been digitalized using 275 ArcGIS® hydrology tools as carried out by other authors (e.g., Pathak et al., 2013). The circular 276 statistic has been used for computing the (vector) mean and median values of glaciers and basins aspect 277 (Fisher, 1993).

278 3.5-6 Uncertainty of measurements

All of the imagery and maps were co-registered in the same coordinate system of WGS 1984 UTM Zone
45N. The Landsat scenes were provided in standard terrain-corrected level (Level 1T) with the use of
ground control points (GCPs) and necessary elevation data (LANDSAT SPPA Team,
2015https://earthexplorer.usgs.gov). The ALOS-08 image used here was orthorectified and corrected for
atmospheric effects as described in Salerno et al. (2012).

284 Concerning the accuracy of the measurements, we refer mainly to the work of Tartari et al. 285 (2008), Salerno et al. (2012), and Salerno et al. (2014a) which address in detail the problem of uncertainty 286 in the morphologicalal-morphometric measurements related to ponds and glaciers obtained from remote sensing imagery, maps and photos. The uncertainty in the measurement of a shape's dimension is 287 288 dependent both upon the Linear Error (LE) and its perimeter. In particular for pondsa-(-(as discussed also by many authors, by Fujita et al. (2009), and Gardelle et al. (2011) in the calculation of LE), only the 289 Linear Resolution Error (LRE) needs to be considered (e.g., Fujita et al., (2009), and Gardelle et al., 290 (2011)). as the co-registration error does not play a key role. For instance, the ponds considered here are 291

292 small, and comparisons are made at the entity level and not at the pixel level. Therefore we did not 293 consider the co-registration error because the comparison was not performed pixel by pixel, at the entity 294 level (pond) (Salerno et al., 2012, Salerno et al., 2015, Thakuri et al., 2015; Wang et al., 2015). The LRE 295 is limited by the resolution of the source data. In the specific study of temporal variations of ponds, Fujita 296 et al. (2009) and Salerno et al. (2012) assumed an error of  $\pm 0.5$  pixels, assuming that on average the lake 297 margin passes through the centers of pixels along its perimeter. The uncertainties in the changes in pond 298 surface area- were derived using a standard error propagation rule, i.e., the root sum of the squares 299 (uncertainty =  $\sqrt{e_1^2 + e_2^2}$ ), where  $e_1$  and  $e_2$  are uncertainties from the first and second scene) of the mapping uncertainty in two scene Salerno et al., 2012; Thakuri et al., 2015). 300

**301 3.6-7 Statistical analysis-**

302 In the short-term inter-annual analysis, tThe degree of correlation among the data was verified through 303 the Pearson correlation coefficient (r) after testing that the quantile-quantile plot of model residuals 304 follows a normal distribution (not shown here) (e.g., Venables and Ripley, 2002). All tests are 305 implemented in the software R (R Development Core Team, 2008) with the significance level at p

The observed changes in the surface area of all the considered ponds are listed in Table 3. In general, all unconnected ponds in the last fifty years (1963-2013) decreased by approximately  $10\pm5\%$  in surface area in the last fifty years (1963-2013), with a significant difference based on the Friedman test (p<0.01). Figure 4d and Table 3 show that, until the 2000s, the ponds had a slight but not significant increasing trend (+7±4%, p>0.05). Since 2000, they have decreased significantly (-1.7±0.6% yr-1, p<0.001 corresponding to -22±18%).

339 As for glaciers, Figure 4c reports the glacier surface area changesglaciers surface differences 340 observed across the SNP (approximately 400 km2) observed by Thakuri et al., 2014. They reported a decrease of  $-13\pm3\%$  from 1963 to 2011. We updated this series to 2013 and found a further-loss of surface 341 342 area of  $(-18\pm3\%)$ . For the glaciers located in the basins with the pondscontaining the selected considered 343 ponds, we tracked changes little bit larger. Their overall surface was  $32.2 \text{ km}^2$  in 1963 and  $25.0 \text{ km}^2$  in 2013, with a decrease of -26±20% (Fig. 4c; Table 3). According to many authors (e.g., Loibl et al., 2014), 344 345 as we observe here, the main losses in area over the last decades in the Himalaya have been observed in 346 smaller glaciers.

347 Once we have having analyzed how climate and glacier surface areas have changed over the last fifty 348 years, we can now attempt to understand the causes that have led to the variations observed in the pond population. Usually and lintuitively, an increase in glacier melt is associated with a decrease in glacier 349 350 surface area, as observed here. However, if this inbound componentthe glacier melt was the most 351 significantpredominant element of the water balance, the ponds would be increased. HoweverWhereas, 352 the ponds have decreased since 2000s; thus, the weaker precipitation observed in recent decades seems to 353 have played a more determining role. Nonetheless, thiese general considerationss analysis is extremely 354 broad because it does not consider.,.. for example, a possible different relationship between pond surface 355 area and the degree of glacier coverage in the basin. Therefore, a deeper analysis has been carried out, as 356 shown in the following, to annually trace track the surface areas of 10 selected ponds from 2000 to 2013.

- 357 **5. Discussion**
- 358

**45.1 Short-term inter-annual analysis: investigation on potential drivers of change-**

359 Considering the wide availability of satellite imagery during the 2000-2013 period, an inter-annual
 analysis has been carried on 10 selected ponds in order to investigate the possible drivers of change. This
 361 was made possible exploiting the continuous series of annual pond surface areas on the one side, and
 362 climatic data from Pyramid station on the other.

**363 5.1.1 Trends ofin pond surface areas**

Table 4 provides the morphometric characteristics of 10 selected ponds. We observe that the median features of these ponds are comparable with the entire pond population (Table 2), highlighting the good representativeness of the selected case studies. Figure S13-S12 shows, for each pond, the annual surface area variations that occurred during the 2000-2013 period. All the selected ponds show a significant (p<0.05) decreasing trend according to what has been observed for the whole pond population during the same period-(Table 3; Fig. 4).

370 5.1.2 Trends of possible drivers of change

The selected These continuous annual series have been compared with some-possible drivers of change
 are: temperature (daily maximum, minimum and mean), precipitation, potential evaporation, and glacier
 melt of the pre-monsoon, monsoon (Fig. 5), and post-monsoon seasons. Pyramid data have been used for
 computing or aggregating these variables. The assumption behind this analysis is that these series can be
 considered representative both along the altitudinal gradient and in the different valleys of the SNP. The
 scarcity of land weather data at these elevations makes licit this assumption, although, at this regard, the
 detected drivers of change will be analyzed in this respect in the last paragraph.

[revised manuscript text omitted]

427 5.2 Long-term inter-annual analysis

An inter-annual analysis has been carried out during from 1963 to 2013 on all 64 considered pond in
 order to investigate 1) which morphological boundary conditions control the pond surface area changes
 and 2) the capability of unconnected ponds to infer on the detected drivers of change also in the past when
 land climatic data did not exist.

**432 5.2.1 Morphological boundary conditions controlling the pond surface area changes 4.5 Change in 433 ponds surface area versus morphological boundary conditions.**

434 We analyzed whether all 64 considered ponds experienced changes in surface area in relation to 435 certain morphological boundary conditions, such as the mean elevation of the basin, the pond surface 436 area, the main three valleys of SNP (Fig. 2a), and the glacier cover. In this case, evaluated the normality 437 of data, we apply the ANOVA test as well as the relevant post-hoc test described above. Figure SI4 shows 438 the surface area changes observed during the 1992-2013 period vs morphological factors. The same 439 analysis has been carried out also on 1963-1992 period reporting decidedly similar results (not shown 440 here). We observe that the pond surface area changes are independent from both elevation, valley, and 441 pond size, whereas significant differences can be observed between ponds with and without glacier cover. 442 In particular, ponds-with-glaciers experienced a lower surface area reduction. This analysis reconfirms 443 that the glacier cover at these altitudes is the main discriminant parameters in the the hydrological cycle 444 of unconnected ponds.

445 We now analysed whether ponds with and without glacier cover within their hydrological basin 446 experienced changes in surface area in relation to certain morphological boundary conditions, such as the 447 aspect or and the elevation of the basin. The two classes has been defined accordingly to the observed 448 threshold of 10%. Hereafter, we define these ponds as ponds without glaciers in the basin (ponds-without-449 glaciers), neglecting in this way relatively small glacier bodies, which could possibly be confused with 450 snowfields. The opposite class is defined as ponds with glaciers in the basin (ponds-with-glaciers). Among ponds-with-glaciers, Table 2 shows that they are characterized by a median glacier coverage of 451 452 19%, oriented toward the east-southeast and relatively steep (31°). The observed changes according to 453 this new classification are reported in Table 3.

In this caseanalysis, we apply the Kruskal-Wallis test as the relevant post-hoc test described above.
Figure 9–7 shows the surface area changes observed during the 1992-2013 period. The changes were

456 independent of both elevation and aspect for ponds-without-glaciers (Fig. 9a7a; Fig. 9e7c), whereas 457 significant differences can be observed for ponds-with-glaciers. Ponds located at higher elevations 458 experienced greater decreases (Fig. 9b7b). In particular, ponds over 5400 m a.s.l. decreased significantly 459 (p<0.01) more than ponds located below 5100 m a.s.l. In terms of aspect, the south-oriented ponds (Fig. 460 9d7d) experienced greater decreases, which was significantly different from southeast (p<0.01) and 461 southwest (p<0.01) orientations.

462 The tracking of pond surface areas provides furthermore important information on precipitation and 463 glacier melt trends in space. Ponds-without-glaciers allows to understand that tThe decline of the 464 precipitation in the SNP since 1992 generally occurroccured homogeneously at all elevations and in all 465 valleys independent of their orientation (Fig. 7a; Fig. 7c). Based on the greater loss of surface area for 466 ponds-with-glaciers at lower elevations, we can infer that glacier melt is actually higher at these 467 elevations, surely due to the effect of higher temperatures (Fig. 7b). Even in valleys oriented in directions 468 other than south, we observe greater losses in surface area for ponds-with-glaciers (Fig. 7d). Small 469 glaciers lying in perpendicular valleys, which are much steeper than the north-south-oriented valleys 470 (following the monsoon direction), are likely melting more due to their small size and higher gravitational 471 stresses (e.g., Bolch et al., 2008; Quincey et al., 2009).

**472 5.2.2 Pond surface areas as proxy of past changes of the hydrological cycle**

**473 Climate reconstruction**

474 To reconstruct the climatic trends before the 1990s1994, we compared the annual and seasonal 475 precipitation and temperature time series recorded at Pyramid station since 1994 (Salerno et al., 2015) 476 with selected regional gridded and reanalysis datasets (Table SI1). Table 1 shows the coefficient of 477 correlation found for these comparisons. Era Interim (r = 0.92, p<0.001) for mean temperature (Fig. 4a) 478 and GPCC (r = 0.92, p<0.001) for precipitation (Fig. 4b) provide the best performance at the annual level. 479 Figure SI5 shows the location of Era Interim and GPCC nodes close to the region of investigation and in 480 particular in relation to the Pyramid station. All these The visual comparisons between gridded/reanalysis 481 and land data are shown-visualized in Figure SHSI6. We observe that precipitation increased significantly 482 until the middle 1990s (+25.6%, p< 0.05, 1970-1995 period), then it started to decrease significantly (-483 23.9%, p< 0.01, 1996-2010 period), as observed by the Pyramid station and described by Salerno et al., 2015. The mean temperature shows-reveals a continuous increasing trend (+0.039 °C yr-1, p< 0.001. 484 485 1979-2013 period) that has accelerated since the early of 1990s.

Furthermore, Table 1 shows the low capability of all the products to correctly simulate monsoon
temperatures and in particular the daily maximum ones. Figure S12a S17a reports visually these
correlations at monthly level for maximum temperature, while Figure S12b-S17b highlights the misfit in
the time between the maximum, mean, and minimum temperature trends during the monsoon period.

**490 Analysis of ponds surface area in the last fifty years**

491 Based on the findings related to the main drivers of changes that have influenced the 10 selected
492 ponds, the overall pond population (all 64 ponds) has been subdivided into two classes defined in relation
493 to the glacier cover (%) in their basins. In 2013, 25 ponds presented a glacier cover > 10% (i.e., 40% of
494 the total ponds), and 39 ponds (i.e., 60% of the total ponds) featured glacier coverages less than this
495 threshold. Hereafter, we define these ponds as ponds without glaciers in the basin (ponds without

496 glaciers), neglecting in this way relatively small glacier bodies, which could possibly be confused with
497 snowfields. The opposite class is defined as ponds with glaciers in the basin (ponds with glaciers).
498 Among ponds with glaciers, Table 2 shows that they are characterized by a median glacier coverage of
499 19%, oriented toward the east southeast and very relatively steep (31°).

500 The observed changes according to this new classification are reported in Table 3. The maps in Figure
501 7-8 show the spatial differences between the two pond classes and comparing compare the relative annual
502 rate of change. Generally, no difference can be observed at valley level, as confirmed by the test applied
503 above (Fig. SI4). It is interesting to visually observe most of the pond-without-glaciers increased in the
504 1963-1992 period, while pond-with-glaciers increased in the 1992-2000 period. QuiteAlmost all the
505 considered ponds decreased during 2000-2013 period.

506 , whereas Figure 8-9 trackes their trends over time. We have already discussed (Fig. 4d) that, in 507 general, all unconnected ponds over the last fifty years have decreased by approximately 10%. 508 Additionally, the presence of glaciers within the pond basins results in divergent trends. The surface area 509 of ponds-without-glaciers strongly decreased (-25±6%, p<0.001)-,,-from 1963 to 2013 (Fig. 9a). strongly 510 decreased (25±6%, p<0.001), whereas, for the same period, In contrast, the surface area of ponds-with-511 glaciers decreased much less (-6 $\pm$ 2%, p<0.05) for the same period (Fig. 9bTable-3). Differences in 512 behavior are also noticeable during the intermediateamong the periods pointed out in Table 3. In this case, 513 we compare the median values of the relative annual rates of change. From 1963 to 1992, ponds-without-514 glaciers increased slightly ( $0.9\pm0.5\%$  yr-1, p<0.1), whereas the other ones remained constant ( $0.0\pm0.1\%$ 515  $yr^{-1}$ ). From 1992 to 2000, ponds-without-glaciers decreased slightly (-1.1±1.9%  $yr^{-1}$ , p>0.1), whereas the other ones increased slightly but significantly ( $+0.7\pm0.5\%$  yr-1, p<0.05). In the most recent period (2000 516 to 2013), both categories decreased, but ponds-without-glaciers decreased more (-2.3 $\pm$ 0.7% yr-1, p<0.001; 517 518 -1.5±0.4% yr-1, p<0.001).

The significance of the divergent trend observed between the two groups has been tested for two
periods (1963-1992 and 1992-2013). Based on a Kruskal-Wallis test, in the first period, pPonds-withoutglaciers featured hadpresented significantly (p<0.01) higher increases than ponds-with-glaciers in the first</li>
period (+13±12%; 0±3%, respectively). Differently, in the second period period period period (-38±6%; -6±2%, respectively).

524 Focusing the attention on Figure 9. this analysis concludes by assessing what we have learned from 525 pond surface areas for the last fifty years. An increase in precipitation occurred until the middle 1990s 526 followed by a decrease until recently recent years. This is shown observing the GPCC precipitation series, 527 but it is also confirmed by the behavior of ponds-without-glaciers (Fig. 9a). With regard to the glacier 528 melt, until the 1990s it was constant. Then, an increase occurred in the early 2000s, while in the recent 529 years a declining was observed (Fig. 9b). This is the trend shown by ponds-with-glaciers. Furthermore, 530 since 1994 the glacier melt, calculated directly from the maximum temperature, which has been recorded 531 by the Pyramid Laboratory, is fully in agreement with the behavior of ponds-with-glaciers. Before 1994 532 suitable maximum temperature cannot be derived from Gridded and Reanalysis products (Table 1 and 533 Fig. SI7), but the ponds are able to point out that the glacier melt in those years has been constant. We 534 observed that Ssimply tracking the glacier surface areas \_\_\_\_\_ did not yield information on the temporal 535 behavior of glacier melt. In this regard, aA decrease in glacier surface area has been identified over the 536 last fifty years (Fig. 4c), but this reduction does not correspond to an increase in glacier melt, as normally 537 expected. As discussed by other authors (Thakuri et al., 2014; Salerno et al., 2015; Wagnon et al., 2013), 538 on the south slopes of Mt. Everest, the weaker precipitation iscould be the main cause of glacier 539 shrinkage. In recent years, glaciers are accumulating less than they were decades ago; thus, their size is 540 declining. In contrast, the tracking of pond surface areas demonstrates that glacier melt did not have a

**541 trend congruent to the glacier shrinkage being influence more to the maximum temperature trend.**

**542 Conclusion**

543 The main contribution provided by this study is to have demonstrated for our case study that surface
544 areas of unconnected ponds could be tracked to detect the behavior of precipitation and glacier melt in
545 remote and barely accessible regions where, even for recent decades, few or no time series exist. Local
546 end peculiar morphological conditions of each pond (possibly enhanced or reduced sediment supply,
547 landslides, groundwater, etc...) could influence the pond surface area. However, the significant
548 relationships found here on a wide pond population demonstrate that these factors are secondary
549 respect to the main components of the hydrological cycle.

Unfortunately, before the 2000s, the availability of high resolution satellite imagery is very limited.
However, with the limited data at our disposal, important information on the evolution of certain main
components of the hydrological cycle at high elevations has been discerned: an increase in precipitation
occurred until the middle 1990s followed by a decrease until recently. Until the 1990s, the glacier melt
was constant. Then, an increase occurred in the early 2000s. In recent years, the declining trend observed
for maximum temperature has reduced the glacier melt.

556 In high-elevation Himalayan areas, unconnected glacial ponds have demonstrated a high sensitivity to 557 climate change. In general, over the last fifty years (1963-2013), unconnected ponds have decreased 558 significantly by approximately  $10\pm5\%$ . We attribute this change to both a drop in precipitation and a 559 decrease in glacier melt caused by a decline in the maximum temperature in the recent years. Evaporation 560 has little effect at these elevations and has remained constant over the last decade, during which the main 561 decline in ponds surface area has been observed.

An increase in precipitation occurred until the middle 1990s followed by a decrease until recently.
With regard to the glacier melt, until the 1990s it was constant. Then, an increase occurred in the early
2000s, while in the recent years a declining. Simply tracking the glacier surface areas did not yield
information on the temporal behavior of glacier melt. A decrease in glacier surface area has been
identified over the last fifty years, attributed by other authors to mainly the observed weaker precipitation.
In contrast, the tracking of pond surface areas demonstrates that glacier melt did not have a trend
congruent to the glacier shrinkage being chiefly influenced more toby the maximum temperature trend.

[revised manuscript text omitted]
| 784 | Song, C., Huang, B., Richards, K., Ke, L., and Phan, V. H.: Accelerated lake expansion on the Tibetan    |
| 785 | Plateau in the 2000s: Induced by glacial melting or other processes?, Water Resour. Res., 50(4), 3170-   |
| 786 | 3186, doi: 10.1002/2013WR014724, 2014.                                                            |
| 787 | R Development Core Team. R: A language and environment for statistical computing. R Foundation for       |
| 788 | Statistical Computing, Vienna, Austria, ISBN 3-900051-07-0, URL http://www.R-project.org, 2008.          |
| 789 | Tachikawa, T., Kaku, M., Iwasaki, A., Gesch, D., Oimoen, M., Zhang, Z., Danielson, J., Krieger, T., Cur- |
| 790 | tis, B., Haase, J., Abrams, M., Crippen, R., and Carabajal, C.: ASTER Global Digital Elevation Model     |
| 791 | ter and the Joint Japan-US ASTER Science Team                                                            |
| 793 | (https://lpdaacaster.cr.usgs.gov/GDEM/Summary GDEM2 validation report final.pdf), 2011.                  |
| 794 | Hengl, T. and Reuter, H.: How accurate and usable is GDEM?, a statistical assessment of GDEM using       |
| 795 | LiDAR data, Geomorphometry, 2, 45–48, 2011.                                                              |
| 796 | LANDSAT SPPA Team: IDEAS – LANDSAT Products Description Document, Telespazio VEGA UK                     |
| 797 | (https://earth.esa.int/documents/101/4/6/9851/LANDSAT Products Description Document.pdf),
2015        |
| 799 | Smol J P Douglas M S V: Crossing the final ecological threshold in high Arctic ponds Proceedings         |
| 800 | of the National Academy of Sciences, 104(30), 12395-12397, doi: 10.1073/pnas.0702777104, 2007.           |
| 801 | Fisher, N. I.: Statistical Analysis of Circular Data, Cambridge University Press, Cambridge, U.K., 1993. |
| 802 | Razali, N. M., Waph, Y. B.: Power comparisons of Shapiro-Wilk, Kolmogorov-Smirnov,                       |
| 803 | Lilliefors and Anderson-Darling tests, J. Stat. Model. Anal., 2(1), 21-33, 2011.                         |
| 804 | Pathak, P. and Whalen, S.: Using Geospatial Techniques to Analyse Landscape Factors                      |
| 805 | Controlling Ionic Composition of Arctic Lakes, Toolik Lake Region, Alaska, in: Geographic                |
| 806 | Information Systems: Concepts, Methodologies, Tools, and Applications, edited by                         |
| 807 | Information Resources Management Association, USA, Volume I, 130-150, doi:                               |
| 808 | 10.4018/978-1-4666-2038-4.ch012, 2013.                                                                   |
| 809 | Willmott, C. and Matsuura, K.: Advantages of the Mean Absolute Error (MAE) over the Root                 |
| 810 | Mean Square Error (RMSE) in assessing average model performance, Clim. Res., 30, 79-82,                  |
| 811 | doi: 10.3354/cr030079, 2005.                                                                      |
| 812 | Kayastha, R. B., Ageta, Y. and Nakawo M.: Positive degree-day factors for ablation on glaciers           |
| 813 | in the Nepalese Himalayas: Case study on Glacier AX010 in Shorong Himal, Nepal,                          |
| 814 | Bulletins of Glaciological Research, 17, 1–10, 2000.                                                     |
| 815 |                                                                                                          |
| 816 |                                                                                                          |
| 817 |                                                                                                          |
|     |                                                                                                          |
| 818 |                                                                                                          |
|     |                                                                                                          |
| 819 |                                                                                                          |
| 820 |                                                                                                          |
| -   |                                                                                                          |
| 821 |                                                                                                          |
| 822 |                                                                                                          |

823 Table 1. Coefficients of correlation between precipitation and temperature time series recorded at
824 Pyramid station for the 1994-2013 period and gridded and reanalysis datasets (pre-monsoon, monsoon,
825 and post-monsoon seasons as the months of February to May, June to September, and October to January,
826 respectively). Bold values are significant with p<0.01.</li>

|                  |              | APHRODITE | GPCC | CRU  | ERA Interim |
|------------------|--------------|-----------|------|------|-------------|
| Precipitation    | annual       | 0.43      | 0.75 | 0.34 | 0.33        |
|                  |              | NCEP CFS  | GHCN | CRU  | ERA Interim |
|                  | pre monsoon  | 0.64      |      |      | 0.81        |
| Minimum          | monsoon      | 0.47      |      |      | 0.72        |
| Temperature      | post monsoon | 0.70      |      |      | 0.65        |
|                  | annual       | 0.72      |      |      | 0.92        |
|                  | pre monsoon  | 0.79      | 0.83 | 0.8  | 0.87        |
| M                | monsoon      | 0.61      | 0.51 | 0.42 | 0.67        |
| Mean lemperature | post monsoon | 0.79      | 0.77 | 0.57 | 0.82        |
|                  | annual       | 0.81      | 0.85 | 0.89 | 0.92        |
|                  | pre monsoon  | 0.83      |      |      | 0.88        |
| Maximum          | monsoon      | 0.54      |      |      | 0.45        |
| Temperature      | post monsoon | 0.82      |      |      | 0.86        |
|                  | annual       | 0.70      |      |      | 0.80        |

Ponds are grouped according to the glacier cover present into each pond basin.

Table 2. General summary of the morphological features of all the 64 considered ponds (data from 2013).

| Topography                                              | Glacier cover <10%
median (range) | Glacier cover >10%
median (range) | All lakes
median (range) |
|---------------------------------------------------------|--------------------------------------|--------------------------------------|-----------------------------|
| Pond elevation (m a s l )                               | 5181(4460-5484)                      | 5159(4505-5477)                      | 5170(4460-5484)             |
| Pond area (10 4 m 2 )             | 0.8(0.1-6.2)                         | 1 3(0 3-56 3)                        | 1.1(0.1-56.3)               |
| Basin area (10 4 m 2 )            | 30(2-430)                            | 130(30-2300)                         | 70(2-2300)                  |
| Basin slope (°)                                         | 25(10-39)                            | 29(23-41)                            | 27(10-41)                   |
| Basin aspect (°)                                        | 163(68-256)                          | 141(94-280)                          | 159(68-280)                 |
| Basin mean elevation (m a.s.l.)                         | 5293(4760-5531)                      | 5400(5119-5945)                      | 5315(4760-5945)             |
| Basin/Pond area ratio (m 2 /m 2 ) | 60(3-485)                            | 67(10-523)                           | 64(3-523)                   |
| Glacier area (%)                                        | 0(0-9)                               | 19(10-61)                            | 0.5(0-61)                   |
| Glacier slope (°)                                       |                                      | 31(21-38)                            | -                           |
| Glacier aspect (°)                                      | -                                    | 166(150-250)                         | -                           |
| Glacier mean elevation (m a s l )                       | _                                    | 5680(5470-7500)                      | -                           |
|                                                         |                                      |                                      |                             |

Table 3. General summary of surface area changes related to all 64 considered ponds surface area
changes from 1963 to 2013. The surface area changes of the glaciers located within the basins are also
reported. For each comparison the uncertainty of measurement is also shown. On the right the cumulative
loss respect to 1963 is reported for eac intermediate period (these data are used for Fig. 8). On the left the
relative annual rate are calculated (these data are used for Fig. 7).

| Period           | Pond surface area change |            | Glacier surface
area change | Period      | iod Pond surface area
change            |           |                | e area |                |     |
|------------------|--------------------------|------------|--------------------------------|-------------|--------------------------------------------|-----------|----------------|--------|----------------|-----|
|                  | Cumulative loss (%)      |            | Cumulative loss (%)            |             | Relative annual rate (% yr -1 ) |           |                |        |                |     |
| Glacier coverage | < 10%                    | > 10%      | All ponds                      | All basins  | Glacier coverage                           | < 10%     | > 10%          |        | All ponds      |     |
| 1963-1992        | +13±12 ·                 | 0 ±3       | +3 ±7                          | 8 ±8        | 1963-1992                                  | 0.9 ±0.5  | · 0.0 ±0.1     |        | +0.5 ±0.3      |     |
| 1963-2000        | $-1 \pm 6$               | $+9\pm2$   | * +7 ±4                        | $-2 \pm 8$  | 1992-2000                                  | -1.1 ±1.9 | $+0.7 \pm 0.5$ | *      | $-0.4 \pm 0.1$ |     |
| 1963-2008        | -4 ±5                    | $+3 \pm 2$ | $+1 \pm 4$                     | -13 ±9 **   | 2000-2008                                  | -0.3 ±1.0 | -1.6±0.6       |        | -0.7 ±0.7      |     |
| 1963-2011        | -7 ±6                    | $0\pm 2$   | -2 ±5                          | -14 ±14 **  | 2008-2011                                  | 0.0 ±2.8  | 0.0 ±1.6       |        | 0.0 ±2.2       |     |
| 1963-2013        | -25 ±6 ***               | -6 ±2      | * -10 ±5 **                    | -26 ±20 **  | 2011-2013                                  | -129 ±4.4 | *** -5.8 ±2.5  | *      | -11±3.5        | **  |
| 1992-2013        | -38 ±6 ***               | -6 ±2      | * -13 ±5 **                    | -34 ±15 *** | 2000-2013                                  | -2.3±0.7  | *** -1.5 ±0.4  | ***    | -1.7 ±0.6      | *** |

843 significance: p<0.001 '\*\*\*'; p<0.01 '\*\*'; p<0.05 '\*'; p<0.1 '.'

[revised manuscript text omitted]

significance: p